# Growth-Promoting Effects of Ten Soil Bacterial Strains on Maize, Tomato, Cucumber, and Pepper Under Greenhouse Conditions

**DOI:** 10.3390/plants14121874

**Published:** 2025-06-18

**Authors:** Jovana Anđelković, Tatjana Mihajilov Krstev, Ivica Dimkić, Nikola Unković, Dalibor Stanković, Nataša Joković

**Affiliations:** 1Department of Biology and Ecology, Faculty of Sciences and Mathematics, University of Niš, Višegradska 33, 18000 Niš, Serbia; jovana.andjelkovic@pmf.edu.rs (J.A.); tatjana.mihajilov-krstev@pmf.edu.rs (T.M.K.); 2Faculty of Biology, University of Belgrade, Studentski Trg 16, 11000 Belgrade, Serbia; ivicad@bio.bg.ac.rs (I.D.); unkovicn@bio.bg.ac.rs (N.U.); 3Faculty of Chemistry, University of Belgrade, Studentski Trg 12–16, 11000 Belgrade, Serbia; dalibors@chem.bg.ac.rs

**Keywords:** identification, characterization, seed inoculation, plant morphological parameters, pigment content, elemental content

## Abstract

Beneficial interactions between plants and bacteria are crucial in agricultural practices, as bacteria can improve soil fertility, promote plant growth, and protect plants from pathogens. This study aimed to molecularly identify and characterize soil bacterial isolates and evaluate their effect on the growth of maize (*Zea mays* L.), tomato (*Solanum lycopersici* L.), cucumber (*Cucumis sativus* L.), and pepper (*Capsicum annuum* L.) under greenhouse conditions. Plant growth parameters, including plant height, root length, and fresh (FW) and dry (DW) weights, were measured. Additionally, pigment extraction and element content analysis using leaves were performed. Among the isolates, the most effective strain in the greenhouse experiment was *Bacillus safensis* SCF6, which significantly enhanced plant height and fresh weight across all tested plants, with the greatest influence observed in maize plant height (439.42 ± 6.42 mm), fresh weight (14.07 ± 0.87 g plant^−1^ FW), and dry weight (1.43 ± 0.17 g plant^−1^ DW) compared to the control (364.67 ± 10.33 mm, 9.20 ± 1.16 g plant^−1^ FW, and 0.92 ± 0.15 g plant^−1^ DW, respectively). Other strains also demonstrated notable results, with *Microbacterium testaceum* SCF4, *Bacillus mojavensis* SCF8, and *Pseudomonas putida* SCF9 showing the highest plant growth-promoting effects on pepper, tomato, and cucumber, respectively. *Pseudomonas putida* SCF9 demonstrated strong antifungal activity against *Monilinia laxa*, with a percentage of mycelial growth inhibition (PGI) of 72.62 ± 2.06%, while *Bacillus subtilis* SCF1 exhibited effects against *Botrytis cinerea* (PGI = 69.57 ± 4.35%) and *Cercospora* sp. (PGI = 63.11 ± 1.12%). The development and application of beneficial bacterial inoculants or their formulated products can contribute to environmentally friendly farming practices and global food security.

## 1. Introduction

One of the most important cereal crops, ranking third in world production, is maize (*Zea mays* L.), which plays a critical role in the food, oil, starch, and alcoholic beverage industries [1]. Tomato (*Solanum lycopersici* L.), cucumber (*Cucumis sativus* L.), and pepper (*Capsicum annuum* L.) are among the most popular vegetable crops consumed for human nutrition, often used fresh in salads or processed into various products like juices, sauces, or salted and pickled vegetables. Due to their high energy content, rich supply of vitamins, dietary fibers, natural acids, and antioxidant compounds, their consumption is not only vital for a balanced diet but also complements their applications in traditional medicine and the cosmetic industry [2].

Plant cultivation requires adding nutrients to the soil because plants have a high demand for elements such as nitrogen, phosphorus, and potassium [3]. Optimal crop production relies on the balance of micronutrients and macronutrients in the soil, as the deficiency of one element cannot be compensated for by the presence of another [4]. In agriculture, it is common to use chemical fertilizers, organic resources, and pesticides to improve crop production. The application of chemical fertilizers enables easier absorption of nutrients essential for plant growth, while pesticides protect plants from phytopathogens [5]. However, the intensive use of chemical fertilizers and pesticides in large-scale crop production can lead to changes in the environment and affect human health through the food and water chains [5]. In addition, intensive application of synthetic fertilizers and pesticides can increase the naturally occurring concentration of toxic metals in the soil to harmful levels [6]. The consequences of environmental pollution, particularly water and soil pollution, caused by modern agricultural practices, have become a global concern. Reducing reliance on chemical fertilizers and excessive pesticide use has led many researchers to explore environmentally friendly alternatives. One eco-friendly solution is biofertilizer technology, which plays an important role in enhancing plant biomass, suppressing plant diseases, and alleviating abiotic stress [7].

Soil bacteria play a crucial role in plant growth by colonizing root surfaces and interacting with plants through various mechanisms, including organic matter degradation, nitrogen fixation, mineral solubilization, and the production of phytohormones [8,9]. Additionally, plant growth-promoting bacteria (PGPB) indirectly benefit plants by producing antimicrobial compounds and siderophores, inducing systemic resistance, and enhancing plant adaptation to stress conditions [10,11]. However, the plant growth-promoting properties of bacteria are highly strain-dependent [12]. Therefore, it is essential to evaluate how specific PGPB strains influence the growth of particular plant species to optimize their application in agriculture. PGPB have been successfully used as bioinoculants to improve growth and yield across various crops, including wheat (*Triticum aestivum* L.) [13], rice (*Oryza sativa* L.) [14], maize [15], tomato [16], cucumber [17], and pepper [18]. Strains such as *Enterobacter cloacae* [19], *Pseudomonas putida* [20], *Bacillus subtilis*, and *Azospirillum brasilense* [21] have demonstrated effectiveness in improving maize germination, root and shoot length, biomass, lateral root development, and yield. The application of a bacterial consortium consisting of *Bacillus pumilus*, *Bacillus amyloliquefaciens*, *Bacillus mojavensis*, and *Pseudomonas putida* at specific developmental stages of tomato plants resulted in significantly greater improvements in plant growth, yield, and nutrient uptake compared to a single inoculation and control [22]. Seed inoculation with *Pseudomonas stutzeri*, *Bacillus subtilis*, *Stenotrophomonas maltophilia*, and *B. amyloliquefaciens* significantly increased germination rates, seedling vigor, growth, and nitrogen content in the root and shoot of cucumber plants compared to untreated plants [23]. Furthermore, *Bacillus* spp. isolated from tomatoes [24] and *P. putida* from maize rhizospheres [25] have shown strong antagonistic effects against phytopathogenic fungi.

Some bacterial strains also tolerate soil contaminated with pesticides or toxic metals and can contribute to their degradation and/or removal, thereby improving soil health and productivity [9]. The resistance of beneficial bacteria in such environments supports their potential application as bioinoculants in sustainable agriculture. It was reported that some toxic metal-resistant PGP bacteria had a mitigating effect on toxic metals in maize plants [26]. Biofortification, a strategy for enhancing the nutritional quality of crops, has traditionally relied on chemical fertilizers. However, the excessive use of these fertilizers may lead to increased accumulation of toxic metals in the soil. Integrating plant–microbe interactions with innovative and sustainable crop production alternatives can reduce dependency on chemical fertilizers, minimize environmental pollution, and lower production costs [27]. Under greenhouse conditions, the *P. putida* strain was found to enhance cucumber quality by improving fruit aroma, flavor, and juice content while simultaneously reducing nitrate and toxic metal levels [28].

Beneficial interactions between plants and bacteria play a significant role in modern agricultural practices, particularly through the use of biofertilizers and biopesticides [29,30]. Biofertilizers contain live or latent beneficial microorganisms formulated in a carrier medium, often supplemented with nutrients and additives to ensure the product’s shelf life [7]. These products are available in solid or liquid formulations and can be applied to soil, seeds, or seedlings to improve plant growth. In the last few decades, biofertilizers have become available worldwide, but nitrogen-fixing biofertilizers still dominate [31]. A major limitation in biofertilizer technology is the reduced shelf life of non-sporulating bacteria. The specific PGP characteristics of bacteria, interactions with other microorganisms under field conditions, the influence of the environment on growth and cell survival, and compatibility with the crop are also important factors in selecting beneficial microorganisms for biofertilizer production [31]. Ensuring product quality and efficacy is crucial for the effective use of biofertilizers and biopesticides. However, it is equally important to assess the potential pathogenicity of PGP microorganisms for humans and the antibiotic resistance profile to avoid the dissemination of antibiotic resistance genes (ARGs) and ensure product safety [32].

The aim of this study was to evaluate the effects of 10 bacterial isolates from the Soil Collection of Fertico—SCF (Fertico doo, Inđija, Serbia) on the initial growth of maize, tomato, cucumber, and pepper under greenhouse conditions. The isolates were molecularly identified by sequencing 16S rRNA, *rpoB* (encodes the β subunit of bacterial RNA polymerase), and *tuf* (encodes the elongation factor EF-Tu) genes and were tested for various plant growth-promoting characteristics, antifungal activity, and sensitivity to antibiotics, toxic metals, and pesticides. By inoculating seeds with individual bacterial strains and cultivating plants in a greenhouse, this research aimed to highlight the potential of SCF bacteria in enhancing plant growth, thereby contributing to sustainable and eco-friendly agricultural practices.

## 2. Results

### 2.1. Molecular Identification of Bacterial Isolates

The best possible identification of the used isolates was achieved by sequencing 16S rRNA, *rpoB,* and *tuf* genes, as is shown in Table 1. Initial identification was performed by 16S rRNA sequencing using 16S ribosomal RNA and core nucleotide databases (Appendix A).

The *rpoB* and *tuf* genes were used to identify *P. putida* and most *Bacillus* isolates when 16S rRNA gene sequencing was insufficient for species-level identification. The isolates were identified by comparing their sequences with the closest reference or other strains in the NCBI GenBank database. They were confirmed to belong to species within the genera *Bacillus*, *Glutamicibacter*, *Microbacterium*, *Pseudomonas*, and *Enterobacter*, with highest sequence similarities ranging from 98.58% to 99.89%. While most bacteria are beneficial and belong to biosafety level 1, *E. cloacae* could be a potentially opportunistic pathogen and belongs to biosafety level 2, according to the French and German risk group classification, which poses a challenge in agricultural practices.

### 2.2. Physiological, Biochemical, and Enzymatic Characteristics of Bacterial Isolates

The results of the physiological, biochemical, and enzymatic characterization of the bacterial isolates are presented in Appendix A. Most isolates exhibited the ability to grow within a temperature range of 4–40 °C, sodium chloride (NaCl) concentrations of 3–7%, and pH values of 5–9. However, *M. testaceum* SCF4, *B. pumilus* SCF7, and *B. mojavensis* SCF8 isolates did not grow at 4 °C, while *G. halophytocola* SCF5 failed to grow at 40 °C. Increased NaCl concentrations were identified as a limiting factor for *M. testaceum* SCF4 and *E. cloacae* SCF10.

The biochemical characterization of the isolates included production of catalase and urease, nitrate reduction, fermentation of sugars, and hydrogen sulfide (H_2_S) and gas production. *B. subtilis* SCF1 and *P. putida* SCF9 were able to reduce nitrate to nitrite, while urease activity was observed only in *B. paralicheniformis* SCF3. Glucose fermentation, along with hydrogen sulfide and gas production, was exhibited by *E. cloacae* SCF10.

Assessment of extracellular enzyme activity (EA) revealed that *B. mojavensis* SCF8 had the highest number of positive results, including the greatest level of mannanase production (EA = 4.11 ± 0.42). Other efficient enzyme producers included *B. paralicheniformis* SCF3 for cellulase (EA = 4.02 ± 0.31), amylase (EA = 3.44 ± 0.48), and pectinase (EA = 2.20 ± 0.26), as well as *P. putida* SCF9 for lipase (EA = 2.05 ± 0.08).

### 2.3. PGP Characteristics and Antifungal Activity of Bacterial Isolates

The plant growth-promoting characteristics of the bacterial isolates were evaluated and are shown in Table 2. Ammonia production was detected for *B. subtilis* SCF1, *B. safensis* SCF6, *P. putida* SCF9, and *E. cloacae* SCF10. Four strains, *M. testaceum* SCF4, *B. safensis* SCF6, *B. pumilus* SCF7, and *E. cloacae* SCF10, demonstrated phosphate solubilization. The purple rings around the colonies of *B. subtilis* SCF1 and *B. subtilis* SCF2 were observed on CAS agar, while an orange ring was noted around *E. cloacae* SCF10, confirming siderophore production. Hydrogen cyanide (HCN) production was detected in *B. safensis* SCF6 and *P. putida* SCF9. Additionally, *P. putida* SCF9 exhibited 1-aminocyclopropane-1-carboxylate deaminase (ACCD) activity, and *B. paralicheniformis* SCF3 was identified as a strong biofilm producer.

The indol-3-acetic acid (IAA) levels were determined during a 7-day incubation period, where six strains (SCF3, SCF4, SCF5, SCF6, SCF8, and SCF10) synthesized IAA. The best producers were *E. cloacae* SCF10 and *M. testaceum* SCF4, which synthesized 83.77 ± 0.65 µg mL^−1^ and 82.71 ± 0.50 µg mL^−1^, respectively.

In the antifungal dual culture method, five isolates (SCF1, SCF2, SCF3, SCF7, and SCF9) inhibited phytopathogens’ growth on PDA plates (Appendix A). The highest percentages of growth inhibition (PGI) were obtained by *P. putida* SCF9 against *Monilinia laxa* (PGI = 72.62 ± 2.06%) and *B. subtilis* SCF1 against *Botrytis cinerea* (PGI = 69.57 ± 4.35%) and *Cercospora* sp. (PGI = 63.11 ± 1.12%) (Figure 1).

### 2.4. Sensitivity of Bacterial Isolates to Antibiotics, Toxic Metals, and Pesticides

The microdilution method was used to determine the sensitivity of bacterial isolates to antibiotics, toxic metals, and pesticides (Table 3). Most isolates exhibited sensitivity to antibiotics, with minimum inhibitory concentrations (MICs) and minimum bactericidal concentrations (MBCs) in the ranges of 0.05–6.25 µg mL^−1^ and 0.05–12.50 µg mL^−1^, respectively. Higher MICs/MBCs concentrations were observed for *B. paralicheniformis* SCF3 (chloramphenicol: MIC = 12.50 µg mL^−1^, MBC = 50.00 µg mL^−1^; penicillin: MIC = 50.00 µg mL^−1^, MBC = 100.00 µg mL^−1^) and for *M. testaceum* SCF4 (ciprofloxacin: MIC = 12.50 µg mL^−1^, MBC = 25.00 µg mL^−1^). Additionally, *E. cloacae* SCF10 exhibited resistance to penicillin and vancomycin.

Toxic metals, such as manganese and lead, influenced the isolates at MICs in the range of 1.95–15.62 mM and MBCs of 1.95–250.00 mM. All isolates were highly sensitive to cadmium, with MICs below the tested concentrations to 0.39 mM and MBCs from 0.02 mM to 6.25 mM, except for *E. cloacae* SCF10 (MIC = 6.25 mM, MBC = 25.00 mM).

The commercial fungicide Blauvit^®^ inhibited the growth of isolates at MICs in the range of 1.00–2.00 mg mL^−1^ and showed a bactericidal effect at MBCs of 2.00–8.00 mg mL^−1^. Equation Pro WG^®^ exhibited MICs and MBCs in the range of 0.60–4.80 mg mL^−1^, while Swich 62.5 WG^®^ showed MICs of 0.20–1.60 mg mL^−1^ and MBCs of 0.80–6.40 mg mL^−1^. Sensitivity to s-metolachlor was observed at MICs ranging from 1.87 to 30.00 mg mL^−1^, with relatively high MBCs of 30.00–60.00 mg mL^−1^. *E. cloacae* SCF10 were resistant to the tested concentrations of Equation Pro WG^®^, Swich 62.5 WG^®^, and herbicides. Isolates showed no sensitivity to the tested concentrations of fluazifop (except SCF2, SCF4, and SCF5) and insecticides.

### 2.5. Effect of Inoculants on Plant Growth

Based on the results obtained, T6 (*B. safensis* inoculant) was the most effective treatment, enhancing morphological parameters across all tested crops. Figure 2 shows some individual plants (both control and treated), where plants from treatment T6 have more developed aerial parts and roots. All measurements are presented in Figure 3.

Treatment T6 significantly improved maize growth, achieving the highest plant height (439.42 ± 6.42 mm), fresh weight (14.07 ± 0.87 g plant^−1^ FW), and dry weight (1.43 ± 0.17 g plant^−1^ DW) compared to the control (364.67 ± 10.33 mm, 9.20±1.16 g plant^−1^ FW, and 0.92 ± 0.15 g plant^−1^ DW, respectively) (Figure 3a,c,d). The greatest root elongation was observed with T10 (185.09 ± 4.19 mm), compared to the untreated plants (154.29 ± 2.89 mm) (Figure 3b). Additionally, treatments T2, T3, T7, T8, T9, and T10 also increased maize plant height and fresh/dry weight, while root length was improved by seven treatments (T2, T3, T4, T5, T6, T8, and T9) (Figure 3).

Three treatments (T3, T6, and T8) significantly enhanced tomato growth, with T8 promoting the greatest plant height (384.96 ± 4.20 mm) compared to untreated plants (353.42 ± 2.95 mm) (Figure 3a). T6 was the only treatment to improve root elongation (145.88 ± 1.65 mm) (Figure 3b), while T3 had the most substantial impact on fresh and dry weight (17.18 ± 1.00 g plant^−1^ and 1.24 ± 0.01 g plant^−1^, respectively), compared to the control (130.75 ± 7.90 mm, 13.27 ± 0.23 g plant^−1^ FW, and 0.88 ± 0.04 g plant^−1^ DW, respectively) (Figure 3c,d).

In cucumber, T9 promoted the best plant height (323.38 ± 6.10 mm, Figure 3a) while T6 also significantly increased plant height and fresh weight (322.96 ± 8.66 mm and 18.03 ± 1.73 g plant^−1^ FW, Figure 3a,c) compared to the control (301.13 ± 2.07 mm and 15.22 ± 0.23 g plant^−1^ FW, respectively). No significant differences in root length and dry weight were observed for cucumber.

Seven treatments (T3, T4, T5, T6, T7, T9, and T10) significantly promoted pepper height (Figure 3a), and most treatments (except T2) enhanced fresh weight (Figure 3c) without affecting root length. The best results were observed with T4, which achieved a pepper plant height (213.92 ± 14.19 mm) and fresh weight (3.91 ± 0.39 g plant^−1^), compared to the control (169.71 ± 8.07 mm and 2.10 ± 0.12 g plant^−1^ FW).

### 2.6. Pigment Content in Leaves

The contents of chlorophyll a (chl a), chlorophyll b (chl b), and carotenoids (cars) were analyzed in fresh leaves of plants treated with different bacterial isolates (Figure 4).

Treatment T1 had the most significant impact on pigment accumulation in maize and pepper, showing the highest levels of chl a (1.38 ± 0.02 mg g^−1^ and 0.62 ± 0.02 mg g^−1^, respectively) and cars (1.07 ± 0.24 mg g^−1^ and 0.30 ± 0.01 mg g^−1^, respectively) compared to the control (Figure 4a,d). Additionally, treatments T3 and T9 increased chl a in pepper, while treatments T2, T3, T4, T5, T9, and T10 significantly enhanced carotenoid levels.

In tomato, five treatments (T5, T6, T8, T9, T10) had the most pronounced effect on chl a accumulation, with T10 exhibiting the highest result (1.10 ± 0.03 mg g^−1^, Figure 4b). Treatment T2 was the only one that significantly enhanced chl a content in cucumber (chl a = 1.00 ± 0.10 mg g^−1^) compared to the control (Figure 4c).

The chlorophyll b content in tomato and cucumber leaves was slightly increased in some treated plants compared to the control, but the differences were not statistically significant. A decrease or complete absence of detectable chl b was observed in maize under treatments T5, T6, T7, and T8. In pepper, chl b was not quantifiable under treatments T1, T6, and T8, and in the control group. The remaining treatments in pepper resulted in an extremely low content of this pigment.

### 2.7. Elemental Content in Leaves

The content of five macroelements (calcium—Ca, potassium—K, magnesium—Mg, phosphorus—P, and sulfur—S) and five microelements (boron—B, copper—Cu, iron—Fe, manganese—Mn, and zinc—Zn) was quantified in dried leaves (DW) using inductively coupled plasma optical emission spectroscopy (ICP-OES) (Table 4).

## 3. Discussion

The soil pollution by chemical fertilizers used to increase crop yield is a worldwide problem. One of the eco-friendly agricultural practices that reduce environmental pollution includes the application of beneficial bacteria as biofertilizers [7]. Bacterial strains in soil interact with plants in various ways, influenced by plant genotypes, root exudates, and external environmental conditions such as climate, the presence of pathogens, and human practices [33]. Although many bacterial strains exhibit plant growth-promoting (PGP) characteristics, it is essential to evaluate their effects on the growth and resistance of specific crops under real conditions.

Isolates from the SCF demonstrated beneficial biochemical and PGP characteristics, suggesting their potential as bioinoculants. They showed good growth under optimal environmental conditions, but *G. halophytocola* SCF5 exhibited growth at an elevated NaCl concentration in the medium. Previous studies have demonstrated that the halotolerant strain *G. halophytocola* improved tomato seedling growth in saline soil [34].

Microorganisms in the rhizosphere can release extracellular enzymes that degrade complex organic polymers (e.g., cellulose, chitin, lignin, and proteins) into smaller molecules, such as simple sugars, amino acids, or organic acids. These molecules are metabolized and released as mineral nutrients, including nitrogen, phosphorus, and sulfur, which are made available for plant uptake [35]. Seven bacteria in the SCF collection demonstrated extracellular enzyme production, and many of them exhibited cellulase activity, a mechanism known to protect plants from phytopathogenic fungi [36]. Isolates capable of producing extracellular enzymes and catalase are expected to exhibit greater resistance to stress conditions, suggesting enhanced environmental resilience, crucial for supporting plant cultivation under stressful conditions and for inducing systemic plant resistance [37]. Additionally, the urease activity and nitrate reduction ability of some isolates in this study indicate their potential to enhance soil quality [38].

The ability of PGP bacteria to increase soil fertility and support plant development is closely associated with their capacity to produce and mobilize essential nutrients for plants. Among these, ammonia production plays a crucial role in providing nitrogen that indirectly stimulates plant development [39]. Additionally, phosphate solubilization is a well-documented mechanism that improves nutrient availability, contributing to increased root length, shoot height, leaf chlorophyll content, and yield parameters in maize [40,41]. The observed difference in phosphate solubilization between the two assays used in this study arises from the inherent characteristics of each method. PVK agar, commonly used for qualitative screening, enables visual detection of clear halo zones around bacterial colonies, facilitating the identification of phosphate-solubilizing bacteria [42]. In contrast, NBRIP medium, with its simplified and defined composition, is more suitable for quantitative analysis in liquid culture, as it reduces background interference and improves measurement accuracy [43]. Phytohormones, particularly IAA, play a critical role in stimulating cell elongation and division, enhancing root development, improving nutrient uptake, and ultimately increasing plant growth [44]. Several strains of *Enterobacter* [45], *Pseudomonas* [46], and *Bacillus* [47] are well-known for their excellent phytohormone production capabilities. Effective colonization of the plant rhizosphere is an important trait, as it enables the bacteria to establish a stable association with plant roots. This process is more efficiently achieved by bacteria capable of forming biofilms, which provide structural protection and facilitate nutrient exchange in the rhizosphere environment [44]. Siderophore production is also a vital trait of PGP bacteria, as these high-affinity compounds inhibit pathogens by chelating or binding Fe^3+^ in the root environment [37]. Similarly, HCN production, as a highly toxic product for fungal cells, makes certain bacterial strains promising candidates for a biocontrol strategy [48]. Another important mechanism is the ACC deaminase activity of bacteria, which regulates the production of ethylene, a stress-related phytohormone, thereby alleviating plant stress and promoting growth under challenging conditions [37].

In addition to promoting plant growth, bacterial strains with antifungal activity can suppress phytopathogenic fungi and control plant disease [36]. Their biocontrol potential is attributed to mechanisms such as antibiotic synthesis, lytic enzymes production, siderophore production, or hydrogen cyanide synthesis [48]. This study demonstrated that several isolates produce metabolites capable of preventing plant diseases caused by *M. laxa*, *B. cinerea*, and *Cercospora* sp., consistent with previous research findings [49,50,51].

The antibiotic sensitivity of potential bioinoculants is crucial to ensure their safety and minimize environmental and health risks [32]. In this study, most isolates demonstrated high sensitivity to low antibiotic concentrations, making them suitable for biofertilizer development. Despite PGP characteristics, the inoculation of crops with bacteria carrying antibiotic resistance genes (ARGs) could negatively impact ecosystems by promoting the spread of antibiotic resistance [32].

The resistance of bacteria to toxic metals and pesticides relies on genetic and physiological adaptations, making them useful for crop production in polluted soils [52]. Increased concentrations of toxic metals and pesticides can negatively impact microbial cell membranes [53,54], cell count [53,54], extracellular enzyme synthesis [54,55], and PGP activity [56,57,58,59], while also disrupting natural bacterial populations [59]. However, strains resistant to elevated toxic metal concentrations can maintain their growth and metabolism, highlighting their potential in phytoremediation [56]. In agreement with the present findings, a previous study also reported that *E. cloacae*, isolated from polluted soil, exhibited high tolerance to cadmium [60]. While copper hydroxide-based fungicides effectively reduce fungal pathogens, they also decrease PGP bacteria populations and increase the presence of potential pathogenic bacteria [61]. In the present study, Blauvit^®^, which contains copper hydroxide, exhibited antibacterial activity at the manufacturer-recommended concentration. Additionally, Switch^®^ and cymoxanil, the active ingredient in Equation Pro WG^®^, have been shown to inhibit PGP bacteria from the Firmicutes and Actinobacteria groups [62]. The results confirmed that six strains were sensitive to the recommended dose of Switch^®^ and highly tolerant to Equation Pro^®^, herbicides, and insecticides. Bacterial strains isolated from the rhizospheres of mustard, chickpea, pea, greengram, and lentil exhibited tolerance to selected herbicides and insecticides at concentrations ranging from 400 to 3200 µg mL^−1^ [63].

In a greenhouse experiment, *B. safensis* SCF6 was the most effective strain, significantly enhancing all morphological parameters in maize and tomato, and improving the plant height and fresh weight in pepper and cucumber. The effects of this isolate on plant growth can be attributed to its ability to synthesize extracellular enzymes (cellulase and lipase), produce IAA, and solubilize phosphorus. These findings align with limited previous studies that demonstrated the potential of *B. safensis* to enhance maize growth under salinity stress [64] and promote the growth of wheat and soybean under normal environmental conditions [65]. Inoculation of tomato with a combination of *B. safensis* and *B. siamensis* improved nutrient uptake, increasing N, P, and K contents in plant tissues [66]. To our knowledge, no studies have yet examined the impact of the *B. safensis* strain on cucumber under greenhouse conditions. Pepper growth was stimulated by most *Bacullus* isolates, but *M. testaceum* SCF4, which exhibited phosphate solubilization and high IAA production, contributed the most to plant development. Previously, *M. testaceum* stimulated the growth of rice seedlings [67] and *Microbacterium. albopurpureum* sp. nov. promoted the development of cucumber and wheat after seed inoculation [68]. The observed effects of *M. testaceum* on pepper growth have not been demonstrated before. *B. mojavensis* SCF8 had the greatest influence on tomato growth, increasing plant height and root length. This strain synthesized extracellular enzymes (mannanase, cellulase, lipase, amylase, and pectinase) and produced IAA. Previous research has reported that *B. mojavensis* seed inoculation enhanced tomato shoot and root length, and fresh weight after 30 days of cultivation [69]. In cucumber, *B. safensis* SCF6 was the most effective in promoting plant height, but *P. putida* SCF9 showed the highest improvement in fresh weight. *P. putida* SCF9 produced extracellular enzymes (mannanase, cellulase, lipase, and amylase) and ammonia, and demonstrated ACC deaminase activity. Additionally, as a producer of HCN and with good antifungal activity, it could contribute to biocontrol. Previous studies have shown that *P. putida* has stimulatory effects on the growth of cucumber [28], tomato [22], chickpea (*Cicer arietinum* L.) [70], and mustard (*Brassica nigra* L.) [71].

Plant growth-promoting bacteria can also indirectly influence plant water relations by enhancing root development, increasing water uptake efficiency, or regulating stomatal activity [72]. In this study, the maize treated by *B. safensis* SCF6 exhibited the highest fresh and dry weights, suggesting efficient water uptake along with improved biomass production and nutrient assimilation. Similar trends were observed in other maize treatments, as well as in tomato treatments *B. paralicheniformis* SCF3 and *B. safensis* SCF6, and in cucumber under *E. cloacae* SCF10, where proportional increases in dry weight accompanied increases in fresh weight. In contrast, tomato treatment with *B. mojavensis* SCF8 and most pepper treatments showed elevated fresh biomass without a corresponding rise in dry biomass. These discrepancies suggest that such treatments primarily enhanced tissue hydration or water retention rather than biomass accumulation, likely through increased cell turgor or transient cell expansion [72,73].

Furthermore, it was confirmed that *B. paralicheniformis* SCF3 and *E. cloacae* SCF10 stimulated maize and pepper growth. However, their high antibiotic resistance and the potential opportunistic pathogenicity of *E. cloacae* make them unsuitable for agricultural application [74]. The promoting effects of *B. paralicheniformis* in the canola (*Brassica napus* L. var. *napus*) rhizosphere community have been previously demonstrated, where it likely enhances plant growth by improving nutrient availability and modulating soil microbial interactions [75]. Similarly, beneficial *Enterobacter* species are well-known for their plant growth-promoting properties, significantly enhancing maize growth through nutrient solubilization and phytohormone production [15,76].

The plant growth-promoting effects of bacteria are highly strain-dependent, with different bacterial strains exhibiting varying direct and indirect mechanisms [12]. The effectiveness of these mechanisms is closely related to the physicochemical properties of the soil, ecological conditions, plant types, and microbial interactions [77]. Consequently, some isolates with favorable PGP characteristics in vitro may not positively impact plant growth under certain conditions. In our study, *B.subtilis* SCF1 did not affect the growth of any of the four selected plant species, whereas *B.subtilis* SCF2 significantly promoted maize growth only. The potential of *B. subtilis* strains as biofertilizers and biocontrol agents for maize cultivation is well established, primarily due to their ability to enhance growth parameters, mitigate stress effects, and protect plants from phytopathogens [78,79]. Otherwise, the PGP activities of eight *B.subtilis* isolates were examined, with BS-263 exhibiting multiple PGP traits but also reducing maize growth under greenhouse conditions [12].

The role of PGP bacteria in enhancing plant pigment levels is examined in crops such as maize [80], rice [81], and tomato [82]. In our study, increased pigment synthesis can be attributed to bacterial siderophores and their ability to improve iron availability [83], as well as the role of synthesized IAA in chloroplast function [84]. The reduced content of chlorophyll b observed in certain maize and pepper treatments may be attributed to oxidative stress, which increases the production of reactive oxygen species (ROS) capable of disrupting enzymes involved in chlorophyll b biosynthesis [85]. Inoculation with PGPB generally enhances a plant’s antioxidant capacity, but in some cases, it may result in elevated levels of oxidative stress markers [86]. Additionally, the method used for pigment extraction plays a crucial role in determining the accuracy of chlorophyll measurements. Variables such as solvent type, temperature, light exposure, and sample preparation techniques can significantly impact pigment stability and yield [87].

Microbe-mediated biofortification, a sustainable agricultural practice, enhances the concentration of essential nutrients, vitamins, and minerals in edible plant parts, contributing to improved food quality and security [27]. In the present study, most treatments exhibited biofortification potential, increasing up to six elements in maize, tomato, cucumber, and pepper leaves. The effectiveness of microbe-mediated biofortification has been confirmed in previous studies, which reported increased trace element concentrations in wheat, red cabbage (*Brassica oleraceae* L. convar. *capitata* (L.) Alef. var. *rubra* (L.) Thell.) [88], and rice [89].

The application of plant growth-promoting bacteria significantly influenced the accumulation of macro- and micronutrients in the leaves of maize, tomato, cucumber, and pepper. The effectiveness of individual bacterial strains varied depending on the plant species, highlighting the specificity of plant–microbe interactions and the diverse plant growth-promoting traits of these isolates, such as phosphate solubilization, siderophore production, indole-3-acetic acid (IAA) synthesis, and ammonia production.

For instance, *M. testaceum* SCF4, *B. safensis* SCF6, *B. pumilus* SCF7, and *E. cloacae* SCF10 contributed to increased phosphorus content in tomato, cucumber, and pepper. This is consistent with earlier findings showing that bioinoculants improve plant nutrient acquisition and growth via phosphate solubilization and IAA synthesis [90]. Similarly, *B. pumilus* has been reported to enhance elemental contents in rice [91], while *B. safensis* improved the uptake of phosphorus, nitrogen, potassium, and zinc in *Stevia rebaudiana* [92]. *E. cloacae* solubilized phosphate and improved nutrient uptake in maize [93]. In our study, several bacterial strains significantly increased the iron content in maize and tomato, likely due to their siderophore-producing capacity. A similar effect has been reported for *B. subtilis*, which improved iron uptake and growth in maize [94]. *B. paralicheniformis* promoted root system development in soybean (*Glycine max* L.) [95], and *G. halophytocola* improved tomato growth [34], by promoting mineral nutrition in plants. Additionally, *Bacillus* spp. strains, capable of solubilizing manganese, phosphorus, and potassium, significantly promoted maize growth and facilitated the mobilization of elements from the soil solution into roots and shoots [96]. However, some treatments resulted in lower element contents compared to the control, highlighting the strain-dependent nature of PGP bacteria and the influence of environmental conditions [77].

## 4. Materials and Methods

### 4.1. Collection and Culturing of Bacterial Isolates

The bacterial isolates were obtained from the Soil Collection of Fertico (Fertico doo, Inđija, Serbia). Ten isolates were taken from stock cultures, transferred onto Luria–Bertani agar (LBA), and incubated at 30 °C for 24 h. The LBA was prepared according to the protocol described previously [97], with the addition of 20 g of agar powder to 1 L of broth. The overnight bacterial cultures were used for isolate characterization and suspension preparation.

For suspension preparation, Luria–Bertani broth (LB broth) was inoculated with the overnight bacterial cultures and incubated at 30 °C and 150 rpm on a rotary shaker (E-20, Biosan, Riga, Latvia) for 24 h. LB broth was prepared following the previously reported protocol [97]. Bacterial cells were then harvested by centrifugation (4000 rpm, 10 min at 4 °C), washed twice with sterile 0.9% sodium chloride, and resuspended in the same solution. The resulting suspensions were adjusted to a turbidity of 0.5 McFarland (approximately 10^8^ colony-forming units per milliliter—CFU mL^−1^) using a densitometer (DEN-1, SIA Biosan, Riga, Latvia) and were used for seed inoculation.

### 4.2. Molecular Identification of Bacterial Isolates

The bacterial isolates were incubated overnight in LB broth, and genomic DNA was extracted using the Zymo Research DNA extraction kit (Irvine, CA, USA) following the manufacturer’s protocol. Molecular identification of the bacterial isolates was performed through partial amplification of the 16S rRNA gene using the primers fD1F-uni-16S and rP2R-uni-16S [98,99]. When the homology of the 16S rRNA gene with the same known genes was not sufficient to determine the species, other conserved genes, such as *rpoB* (encodes the β subunit of bacterial RNA polymerase) and *tuf* (encodes the elongation factor EF-Tu), were amplified and sequenced. For more precise identification of *Bacillus* species, the *tuf* gene was amplified using primers tufGPF and tufGPR [100]. Additional characterization of the other isolates was performed by amplifying the *rpoB* gene using primers rpoB-F4 (5′-AARATGGGCVGGYCGTCACGG-3′) and rpoB-R3 (5′-CCGAARCGCTGYCCACCGAA-3′) (this study). All PCR amplifications were performed in a total reaction volume of 25 µL that consisted of 12.5 µL of FastGene Taq 2× Ready Mix (NIPPON Genetics, Tokyo, Japan), 9.5 µL of PCR DNase/RNase free water, 1 µL of each of the primers (10 µM), and 1 µL of sample DNA. 16S rRNA PCR was performed with the HotStarTaq Plus Master Mix Kit (Qiagen, Germantown, MD, USA) under the following conditions: 94 °C for 3 min, followed by 30 cycles of 94 °C for 30 s, 54 °C for 45 s, and 72 °C for 1 min, with a final extension cycle at 72 °C for 5 min. The other *tuf* and *rpoB* PCRs were performed under similar conditions, with 35 cycles and annealing temperature at 55 and 54 °C, respectively. The expected PCR product sizes for 16S rRNA, *tuf*, and *rpoB* were 1500 bp, 791 bp, and 600–700 bp, respectively. The resulting PCR products were purified using the DNA Clean & Concentrator and Zymoclean Gel DNA Recovery kit (Zymo Research, Irvine, CA, USA) before being sent for sequencing (Eurofins, Ebersberg, Germany). Forward primers were used for sequencing, except for the 16S rRNA, where primer 907R-16S was exploited [101]. The obtained sequences were manually checked and aligned, using the ClustalW function of the BioEdit (v.7.2) program, to the ones available in the NCBI database using the BLASTn RefSeq Representative Genomes function (https://blast.ncbi.nlm.nih.gov/Blast.cgi?PROGRAM=blastn&PAGE_TYPE=BlastSearch&LINK_LOC=blasthome, accessed on 7 March 2025).

### 4.3. Characterization of Bacterial Isolates

#### 4.3.1. Physiological Characteristics

The bacterial isolates were examined for their ability to grow under different temperature ranges, salt concentrations, and pH levels on solid media in Petri dishes. To test the influence of temperature, the isolates were grown on LBA at 4 °C, 18 °C, 28 °C, and 40 °C for 24 h. Salt tolerance was assessed on LBA supplemented with sodium chloride (NaCl) at various concentrations: 3%, 5%, 7%, and 9% (*w*/*v*). The analysis of the effect of pH was performed on LBA plates with the pH adjusted to 4, 5, 6, 7, 8, and 9 by adding 1N NaOH or HCl. Bacterial suspensions were spread on the agar plates, and growth was evaluated after 24 h of incubation at 30 °C for salt tolerance and pH influence. Bacterial growth was scored as follows: (−) no growth, (±) low growth, and (+) good growth.

#### 4.3.2. Biochemical Characteristics and Enzyme Production

All bacterial isolates were screened for extracellular enzyme production using plate assays specific to each enzyme. The following enzymes were assessed: cellulase [102], mannanase [103], lipase [104], amylase [105], and pectinase [106]. For each assay, previously prepared bacterial suspensions (Section 4.1) were transferred onto selective agar media containing the appropriate substrate: carboxymethyl cellulose for cellulase, locust bean gum for mannanase, tributyrin for lipase, starch for amylase, and pectin for pectinase. Hydrolytic activity was determined by the presence of halo zones around the bacterial colonies after 48 h of incubation at 30 °C. The enzyme activity (EA) was calculated using the formula [107]$$EA=\frac{diameter\;of\;colony+diameter\;of\;halo\;zone}{diameter\;of\;colony}$$

Biochemical tests were performed following standard methods. Catalase activity was determined by the formation of oxygen bubbles upon the addition of 3% hydrogen peroxide to a bacterial culture, following the previously described method [104]. Urease activity was assessed using Christensen’s urea agar (Himedia, Mumbai, India), where a color change from orange to pink indicated a positive result. Nitrate reduction was evaluated by adding 1 mL of sulfanilic acid solution and 0.5 mL of α-naphthylamine solution to the culture [108]. A positive result was indicated by the development of a red to reddish-brown color in the medium. Citrate utilization was tested using Simmons citrate agar (SCA, Himedia, India), where growth and a color change from green to blue signified a positive result. Fermentation of glucose/lactose, as well as H_2_S, and gas production were tested using Kligler agar (KA, Himedia, India). Color changes in the slant or butt, black precipitate formation, and gas bubbles were indicative of positive metabolic activity. All cultures were incubated at 30 °C, and results were recorded after 24 h of incubation, except for nitrate reduction, which was assessed after 7 days. Results were expressed as follows: (+) for production and (−) for no production.

#### 4.3.3. Plant Growth-Promotion Characteristics

Ammonia (NH_3_) production was tested by culturing bacteria in peptone water for 48–72 h at 30 °C. Subsequently, 0.5 mL of Nessler’s reagent was added to each tube. The development of a brown or yellow color indicated ammonia production [109].

Phosphate solubilization was performed on Pikovskaya agar (PVK) [42] and National Botanical Research Institute’s phosphate agar (NBRIP) [43]. The bacterial isolates were spot-inoculated onto the media and incubated at 30 °C for 14 days. The diameter of clear halo zones around the colony was measured, and the solubilization index (SI) was calculated as the ratio of the total diameter to the colony diameter [110].

The ability of isolates to produce siderophores was tested on Chrom Azurol S agar medium (CAS agar, Acros Organics, Geel, Belgium) which detects iron-chelating activity. The isolates were spot-inoculated onto CAS agar plates and incubated at 30 °C. The appearance of orange or purple rings around the colonies was recorded as a positive result [111].

Hydrocyanic acid (HCN) production was performed according to a previously described method [112]. Bacterial cultures were grown on glycine-supplemented nutrient agar plates. A filter paper strip impregnated with a solution of picric acid and sodium carbonate was placed in the lid of each plate. After incubating the plates at 30 °C for 3–5 days, a color change of the filter paper from yellow to brown or red indicated HCN production.

Screening for 1-aminocyclopropane-1-carboxylate deaminase (ACCD) activity was based on the ability of isolates to utilize ACC (1-aminocyclopropane-1-carboxylate) as the sole nitrogen source. This was conducted on Dworkin and Foster (DF) minimal salts agar supplemented with 3 mM ACC [113]. DF minimal salt agar supplemented with ammonium sulfate was positive control [114].

Biofilm production was assessed using the crystal violet (CV) staining method in microtiter plates [115]. Bacterial cultures were incubated in Mueller–Hinton broth (MHB, Himedia, India) supplemented with 0.5% glucose (in final concentration) at 30 °C for 24 h. After incubation, the wells were washed with phosphate-buffered saline (PBS) to remove planktonic cells and stained with 0.1% CV for 20 min. Stained plates were rinsed off, and each well was filled with 95% ethanol. After 45 min, 100 µL of the solution was transferred and the absorbance was measured at 595 nm using an ELISA reader (Multiscan Ascent, Labsystems, Helsinki, Finland). Results were categorized as non-producer, weak producer, moderate producer, or strong producer [116].

Detection of indole-3-acetic acid (IAA) was performed using a colorimetric method with the Salkowski reagent [117]. LB broth supplemented with 5 mM tryptophan (at the final concentration) was inoculated and incubated for 7 days on a rotary shaker (ES-20, Biosan, Riga, Latvia) at 150 rpm and 30 °C. Each day, a sample was centrifuged at 12,000 rpm and 2 mL of Salkowski reagent was added to 1 mL of the supernatant. After 25 min of incubation at room temperature, absorbance was measured at 530 nm using a spectrophotometer (UV/VIS 1650 PC, Shimadzu, Kyoto, Japan). The amount of synthesized IAA in the supernatant was determined using the standard curve with different IAA concentrations (0, 5, 10, 20, 35, 50, and 100 µg mL^−1^) (Sigma-Aldrich, Darmstadt, Germany).

#### 4.3.4. Antifungal Activity

The antifungal activity of bacterial isolates was tested against ten phytopathogenic fungi: *Botrytis cinerea*, *Rhizoctonia solani*, *Monilinia laxa*, *Fusarium graminearum*, *Fusarium boothii*, *Fusarium oxysporum*, *Alternaria alstroemeriae*, *Aspergillus flavus*, *Cercospora* sp., and *Didymella keratinophila* (collection of the University of Belgrade, Faculty of Biology, Belgrade). The assay was performed using the dual culture method in Petri dishes [118]. A mycelial plug was transferred to potato dextrose agar (PDA, Himedia, India) and positioned 2 cm from the edge of the dish. For the treatment (T), 10 µL of the bacterial suspension was streaked 2 cm from the edge on the opposite side of the plate. For the control (C), sterile water was streaked instead of the bacterial suspension. After 7 days of incubation at 28 °C, the mycelial growth on both treatment and control plates was measured in millimeters (mm). The percentage of growth inhibition (PGI) caused by the bacterial isolate was calculated using the following formula:$$PGI\left(\%\right)=\frac{C-T}{C}\times 100$$
where C is mycelial growth on the control plate (in mm) and T is mycelial growth on the treatment plate (in mm).

#### 4.3.5. Sensitivity of Bacterial Isolates to Antibiotics, Toxic Metals, and Pesticides

The determination of the minimum inhibitory concentration (MIC) and minimum bactericidal concentration (MBC) of tested agents (antibiotics, toxic metals, and pesticides) against selected bacterial isolates was performed according to the Micro-well Dilution Assay [119]. Antibiotics—chloramphenicol, penicillin, ciprofloxacin, and vancomycin (Sigma-Aldrich, Darmstadt, Germany)—were tested at concentration ranges of 0.05–100.00 µg mL^−1^. Toxic metals, including MnCl_2_ × 4H_2_O and PbCl_2_ (Sigma, Germany), were tested in the ranges of 0.20–500.00 mM, while CdCl_2_ (Sigma, Germany) was tested at concentrations of 0.02–50.00 mM. The tested concentrations for commercial pesticides were as follows: 0.06–128.00 mg mL^−1^ for Blauvit^®^ (Župa, Kruševac, Serbia), 0.01–19.20 mg mL^−1^ for Equation Pro WG^®^ (DuPont, Geneva, Switzerland), and 0.01–25.60 mg mL^−1^ for Swich 62.5 WG^®^ (Syngenta, Basel, Switzerland). Herbicides were tested at concentrations of 0.03–60.00 mg mL^−1^ for s-metolachlor (Sigma, Germany) and of 0.01–20.00 mg mL^−1^ for fluazifop p-butil (Sigma, Germany). Insecticides, including deltamethrin and cypermethrin (Sigma, Germany), were tested at concentrations in the ranges of 0.0002–0.40 mg mL^−1^ and 0.002–3.20 mg mL^−1^, respectively. The range of tested concentrations for pesticides was calculated based on the manufacturers’ recommended doses for commercial fungicides or the active ingredient content (s-metolachlor, fluazifop p-butil, deltamethrin, and cypermethrin) in commercial herbicides and insecticides. All agents were initially dissolved in Mueller–Hinton broth (MHB), except for herbicides and insecticides, which were prepared in 100% dimethyl sulfoxide (DMSO). A serial two-fold dilution was performed, with a bacterial cell concentration of 10^6^ CFUmL^−1^. Microplates were incubated at 30 °C for 24 h, and all determinations were performed in triplicate. The MIC was defined as the lowest concentration with no visible growth in MHB, while MBC was defined as the lowest concentration at which no colony growth was observed on Mueller–Hinton agar (MHA, Himedia, Mumbai, India).

### 4.4. Greenhouse Experiment

#### 4.4.1. Preparation and Inoculation of Seeds

Seeds of maize (*Zea mays* L.—ZP 388, Zemun Polje, Serbia), tomato (*Solanum lycopersici* L—Kazanova, Superior, Velika Plana, Serbia), cucumber (*Cucumis sativus* L.—Dugi zeleni, Bioprodukt, Čačak, Serbia), and pepper (*Capsicum annuum* L.—Dukat, Bioprodukt, Čačak, Serbia) were disinfected by soaking in 80% (*v*/*v*) ethanol, followed by sodium hypochlorite (4% active chloride). The seeds were then rinsed three times with sterile water and dried at room temperature [120].

Bacterial suspensions at a concentration of 10^8^ CFUmL^−1^ were prepared, as previously described (Section 4.1). For seed inoculation, the 360 sterilized seeds of each plant species (maize, tomato, cucumber, and pepper) were soaked separately in 1 mL of the appropriate bacterial suspension (12 seeds of each plant species per treatment in three replicates), for 30 min. These inoculated seeds were then used in the greenhouse experiment. The experimental design consisted of 10 treatment groups, each inoculated with a specific bacterial isolate, labeled as follows: T1—*Bacillus subtilis* SCF1, T2—*Bacillus subtilis* SCF2, T3—*Bacillus paralicheniformis* SCF3, T4—*Microbacterium testaceum* SCF4, T5—*Glutamicibacter halophytocola* SCF5, T6—*Bacillus safensis* SCF6, T7—*Bacillus pumilus* SCF7, T8—*Bacillus mojavensis* SCF8, T9—*Pseudomonas putida* SCF9 and T10—*Enterobacter cloacae* SCF10. The control group (C) consisted of seeds soaked in 0.9% NaCl solution.

#### 4.4.2. Experimental Design

The effect of 10 bacterial isolates on the initial growth of plants was evaluated under greenhouse conditions. The experiment was conducted using plastic trays (dimensions: 54 × 28 cm) containing 72 cells, each with a depth of 4 cm and a single-cell volume of 40 mL. Each cell was filled to the top with sterile commercial substrate (Floragard, Oldenburg, Germany), which had been autoclaved at 121 °C prior to use. For each treatment group (T1–T10), twelve seeds of maize, tomato, cucumber, and sweet pepper, inoculated with their respective bacterial isolates, were sown at a depth of 2 cm in the trays. The seeds were uniformly watered with the same volume of water. The trays were initially placed in a growth chamber set to a constant temperature of 26 °C with a 16 h light and 8 h dark photoperiod until germination occurred.

After germination, the seedlings were transferred to a greenhouse and grown under temperature conditions ranging from 18 to 25 °C. Each of the 10 treatments and the control group were performed in triplicate, resulting in a total of 1584 plants. After 35 days of growth in the greenhouse, the plants were harvested, and their roots were washed with distilled water.

#### 4.4.3. Morphological Parameters

Plants from the treatment and control groups were assessed for various growth parameters, including plant height (measured from the base to the apex) and root length, both recorded in millimeters (mm) using graph paper. All plant samples were then dried at 65 °C for 72 h for further analysis. Additionally, the fresh weight (FW) and dry weight (DW) of the plants were measured and recorded in grams per plant (g plant^−1^).

#### 4.4.4. Pigment Content Quantification Method

Fresh leaves (0.5 g) were chopped and macerated in 80% acetone with the addition of SiO_2_ and CaCO_3_. The resulting macerate was filtered with a vacuum pump, centrifuged at 4000 rpm and 4 °C, and the supernatant was filled up with acetone to the final volume of 25 mL [87]. Each sample was diluted 5-fold, and absorbance was measured using a spectrophotometer (UV/VIS 1650 PC, Shimadzu, Kyoto, Japan) at the following wavelengths: chlorophyll a (chl a) at 663.2 nm, chlorophyll b (chl b) at 646.8 nm, and carotenoids (cars) at 470 nm. The pigment content (in mg mL^−1^) was calculated using the following equations [121]:Chlorophyll a (chl a)=(12.25×A 663.2)−(2.79×A 646.8)$$Chlorophyll\;b\left(chl\;b\right)=\left(21.50\times A\;646.8\right)-(5.10\times A\;663.2)$$$$Carotenoids\left(cars\right)=\frac{\left(\left(1000\times A\;470\right)-1.82\times chl\;a\right)-(85.02\times chl\;b))}{198}$$

The results were expressed as milligrams of pigments per gram of fresh weight (mg g^−1^ FW), following previously described protocols [122].

#### 4.4.5. Elemental Content Analysis

The contents of macroelements (Ca, K, Mg, P and S, expressed in mg g^−1^) and microelements (B, Cu, Fe, Mn and Zn, expressed in µg g^−1^) were determined in dried (DW) and ground leaf samples. Sample preparation was carried out through acid digestion using a microwave oven (Ethos 1, Advanced Microwave Digestion System, Milestone, Sorisole, BG, Italy) as previously described [123]. The analysis was performed with an inductively coupled plasma optical emission spectrometer (ICP-OES 6500 Duo, Thermo Scientific, Cambridge, UK).

### 4.5. Statistical Analysis of Data

The data was analyzed by SPSS software (IBM SPSS Statistics, Version 19.0). A one-way analysis of variance (ANOVA) was performed, followed by Tukey’s post hoc test to assess differences among treatments. Results are expressed as means ± standard deviations (SDs), with statistical significance defined if *p* ˂ 0.05.

## 5. Conclusions

The overuse of chemical fertilizers contributes to environmental degradation and the emergence of phytopathogens. Developing eco-friendly solutions for sustainable agriculture is crucial and involves the use of biofertilizers to promote plant growth and improve soil health. Since the effectiveness of biofertilizers depends on the specific characteristics of microbial strains, it is essential to examine each strain individually to determine its potential benefits and optimal applications. SCF isolates, especially *M. testaceum* SCF4, *B. safensis* SCF6, *B. mojavensis* SCF8, and *P. putida* SCF9, demonstrated the greatest efficiency in enhancing the plant growth. *B. safensis* SCF6 showed the most promising results across all tested crops with the best results in maize seeds inoculation. *M. testaceum* SCF4, *B. mojavensis* SCF8, and *P. putida* SCF9 were identified as the most effective plant growth-promoting bacteria for pepper, tomato, and cucumber, respectively. An in vitro antifungal assay demonstrated strong inhibition of some phytopathogenic mycelial growth by *B. subtilis* SCF1 and *P. putida* SCF9. The selection of plant growth-promoting bacteria for biofertilizers must prioritize their safety to prevent environmental and health issues. Consequently, the application of antibiotic resistant isolates such as *B. paralicheniformis* SCF3 and *E. cloacae* SCF10 is restricted due to the potential risk of spreading antibiotic resistance genes among phytopathogens. Overall, this study highlights the important roles of bacterial isolates in promoting plant growth, enhancing nutrient content, and exhibiting biocontrol potential. Future studies should focus on evaluating the PGP effects of isolates under field conditions. Additionally, exploring combinations of beneficial bacteria from the SCF collection could lead to development of optimized formulations that enhance crop productivity and contribute to sustainable agricultural practices.

## Figures and Tables

**Figure 1 plants-14-01874-f001:**
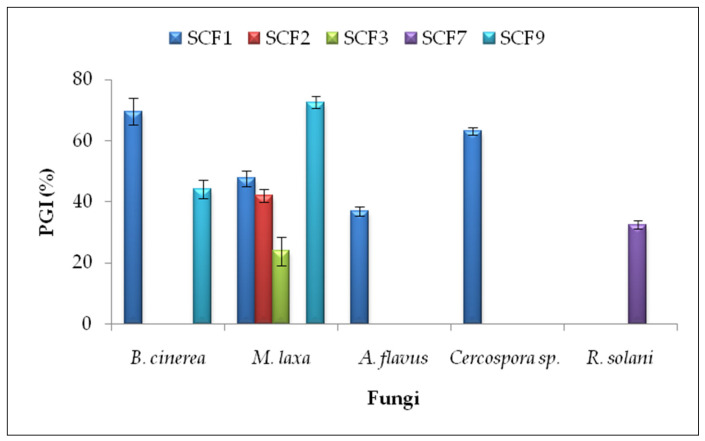
Percentage of mycelial growth inhibition (PGI) in the dual culture method. Results are presented as the means ± standard deviations from three replicates. Isolates that did not inhibit the mycelial growth of phytopathogenic fungi are not included. SCF1—*B. subtilis*; SCF2—*B. subtilis;* SCF3—*B. paralicheniformis*; SCF7—*B. pumilus*; SCF9—*P. putida*.

**Figure 2 plants-14-01874-f002:**
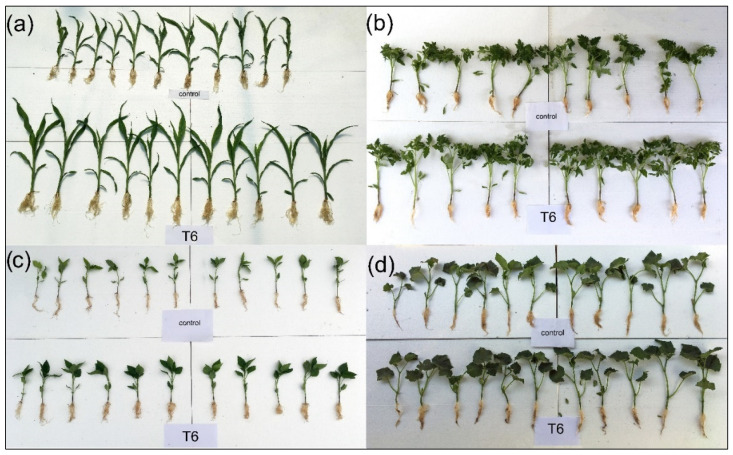
Control plants and plants treated with *B. safensis* SCF6 (treatment T6) after 35 days of growth: (**a**) maize, (**b**) tomato, (**c**) pepper, and (**d**) cucumber.

**Figure 3 plants-14-01874-f003:**
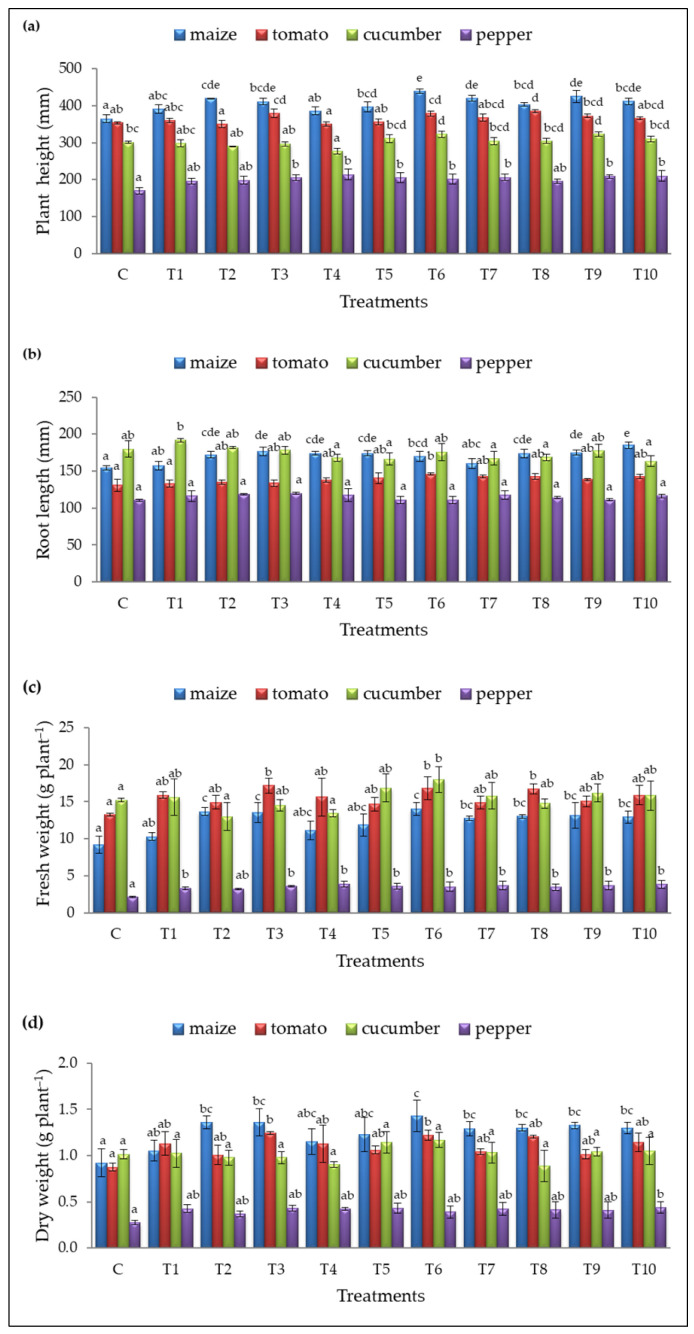
(**a**) Plant height (mm), (**b**) root length (mm), (**c**) fresh weight (FW, in gram per plant), and (**d**) dry weight (DW, in gram per plant) for the control (C) and treatments (T1–T10). Results are expressed as means ± standard deviations from three replicates. Different letters indicate statistically significant differences among treatments (ANOVA, Tukey’s test, *p* ˂ 0.05).

**Figure 4 plants-14-01874-f004:**
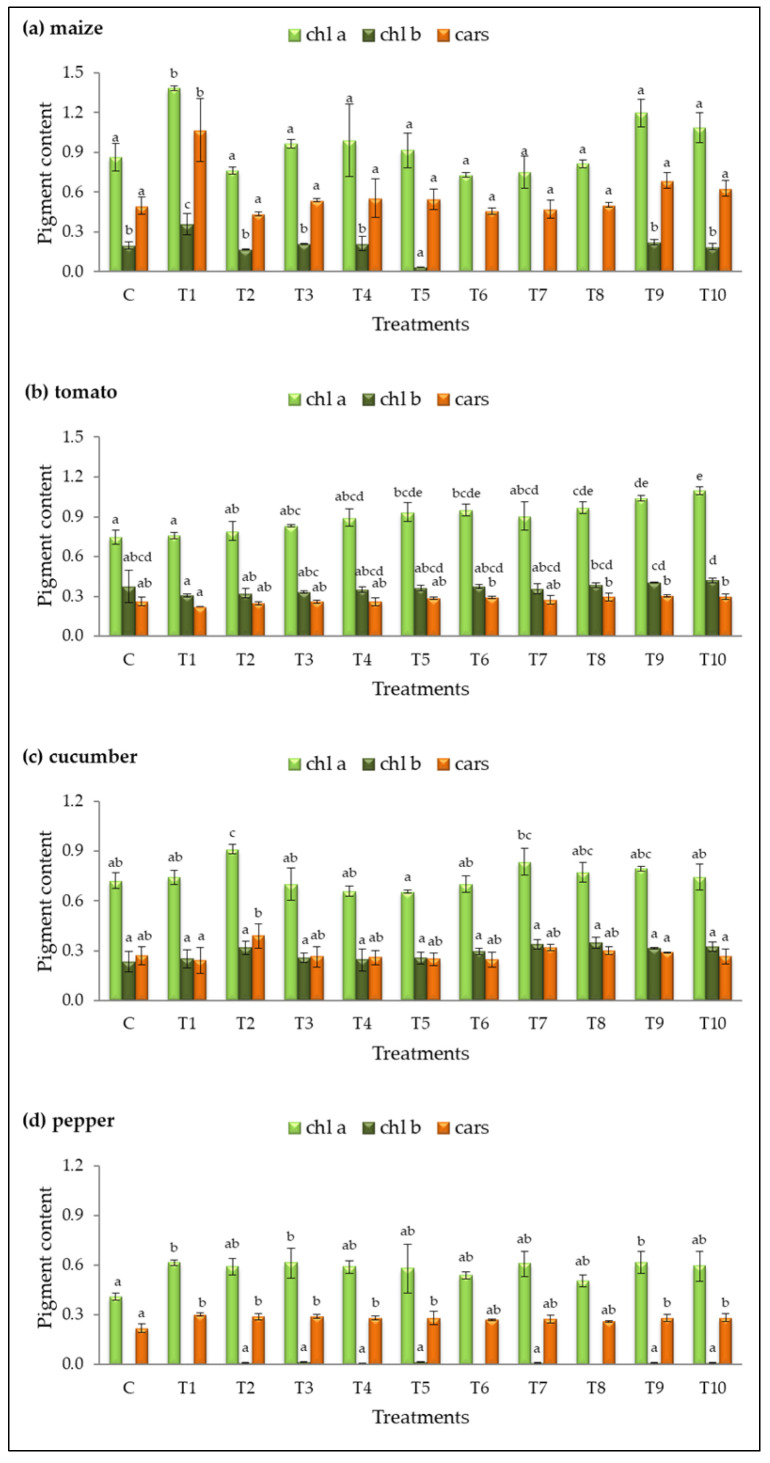
Pigment content (mg∙g^−1^ FW) in fresh leaves of (**a**) maize, (**b**) tomato, (**c**) cucumber, and (**d**) pepper for the control (C) and treatments (T1–T10). Results are expressed as means ± standard deviations from three replicates. Different letters indicate statistically significant differences among treatments (ANOVA, Tukey’s test, *p* ˂ 0.05).

**Table 1 plants-14-01874-t001:** Bacterial isolates from soil collection (SCF) identified by molecular methods.

Isolates	PCR Primers	Homology to the Closest Reference and Other Strains	Similarity	Accession No.
SCF1	tuf-GPF	*Bacillus subtilis* FDAARGOS_606	99.73%	CP041015
SCF2	907R-16S	*Bacillus subtilis* DSM 10	99.87%	NR027552
SCF3	tuf-GPF	*Bacillus paralicheniformis* FA6	99.87%	CP033198
SCF4	907R-16S	*Microbacterium testaceum* DSM 20166	98.81%	NR026163
SCF5	907R-16S	*Glutamicibacter halophytocola* KLBMP 5180	98.58%	NR156872
SCF6	tuf-GPF	*Bacillus safensis* U17-1	99.86%	CP015611
SCF7	tuf-GPF	*Bacillus pumilus* 3–19	99.73%	CP054310
SCF8	907R-16S	*Bacillus mojavensis* IFO15718	99.89%	NR024693
SCF9	rpoB-F4	*Pseudomonas putida* E46	99.46%	CP024086
SCF10	907R-16S	*Enterobacter cloacae* subsp. *dissolvens* ATCC 23373	99.77%	NR118011

**Table 2 plants-14-01874-t002:** Plant growth-promoting (PGP) characteristics and indole-3-acetic acid (IAA) production of SCF isolates.

	Isolates									
	SCF1	SCF2	SCF3	SCF4	SCF5	SCF6	SCF7	SCF8	SCF9	SCF10
PGP										
NH_3_	+	−	−	−	−	+	−	−	+	+
PVK (SI) *	−	−	−	1.57 ±0.21	−	1.89 ±0.35	1.62 ±0.21	−	−	1.87 ±0.32
NBRIP (SI) *	−	−	−	−	−	−	−	−	−	1.35 ±0.02
sider	+/p	+/p	−	−	−	−	−	−	−	+/o
HCN	−	−	−	−	−	+	−	−	+	−
ACCD	−	−	−	−	−	−	−	−	+	−
biofilm	weak	weak	strong	weak	weak	weak	weak	weak	weak	moderate
IAA (µg mL^−1^) *										
1 day	–	–	2.20 ±0.20	24.28 ±0.15	0.80 ±0.05	–	–	–	–	43.34 ±0.45
2 days	–	–	1.65 ±0.16	64.56 ±0.90	1.74 ±0.21	2.27 ±0.34	–	2.56 ±0.19	–	74.80 ±1.30
3 days	–	–	0.77 ±0.10	81.00 ±0.96	2.87 ±0.26	2.56 ±0.39	–	2.31 ±0.12	–	83.77 ±0.65
4 days	–	–	0.80 ±0.10	80.67 ±0.89	2.89 ±0.23	3.33 ±0.14	–	3.83 ±0.12	–	81.10 ±1.30
5 days	–	–	0.68 ±0.06	82.71 ±0.50	3.29 ±0.16	3.84 ±0.51	–	4.29 ±0.14	–	77.49 ±0.42
6 days	–	–	1.26 ±0.30	77.22 ±1.10	3.42 ±0.05	4.47 ±0.17	–	4.86 ±0.30	–	72.44 ±0.30
7 days	–	–	2.33 ±0.17	73.25 ±0.93	3.27 ±0.08	4.58 ±0.18	–	4.76 ±0.13	–	68.35 ±1.28

(−) No production; (+) production; NH_3_—ammonia; PVK—Pikovskaya’s agar; NBRIP—National Botanical Research Institute’s phosphate agar; SI—solubilization index; *—means ± standard deviations; sider—siderophores; p—purple zone color; o—orange zone color; ACCD—1-aminocyclopropane-1-carboxylate deaminase; IAA—indole-3-acetic acid (production during the 7-day incubation period).

**Table 3 plants-14-01874-t003:** Sensitivity of SCF bacterial isolates to antibiotics, toxic metals, and pesticides (commercial fungicides, herbicides, and insecticides).

	Isolates									
	SCF1	SCF2	SCF3	SCF4	SCF5	SCF6	SCF7	SCF8	SCF9	SCF10
	MIC/MBC									
Antibiotics (µg mL^−1^)										
Chl	3.12/3.12	3.12/3.12	12.5/50.00	0.39/1.56	3.12/3.12	3.12/6.25	6.25/6.25	6.25/12.50	1.56/3.12	0.39/25.00
Pen	<0.05/<0.05	<0.05/<0.05	50.00/100.00	<0.05/<0.05	0.05/0.05	<0.05/<0.05	<0.05/0.39	<0.05/<0.05	<0.05/<0.05	>100.00/>100.00
Cip	<0.05/<0.05	1.56/1.56	<0.05/<0.05	12.50/25.00	0.78/0.78	0.10/0.39	0.10/0.39	0.19/0.19	0.10/0.19	0.19/0.19
Van	0.10/0.19	0.39/0.39	0.19/3.12	0.19/0.39	0.19/0.19	0.19/3.12	0.10/1.56	0.19/3.12	0.10/0.19	>100.00/>100.00
Toxic metals (mM)										
Mn	7.81/15.62	3.90/31.25	7.81/125.00	7.81/7.81	1.95/1.95	7.81/125.00	15.62/250.00	7.81/125.00	7.81/7.81	15.62/125.00
Pb	15.62/62.50	15.62/15.62	7.81/15.62	3.90/125.00	1.95/15.62	15.62/62.50	7.81/125.00	15.62/62.50	15.62/15.62	15.62/31.25
Cd	<0.02/3.12	<0.02/0.02	<0.02/0.10	<0.02/1.56	<0.02/3.12	<0.02/3.12	0.10/0.78	0.10/6.25	0.39/0.78	6.25/25.00
Fungicides (mg mL^−1^)										
Bl	2.00/2.00	1.00/2.00	2.00/2.00	1.00/4.00	1.00/2.00	2.00/4.00	2.00/4.00	2.00/4.00	2.00/2.00	2.00/8.00
Eq	1.20/2.40	4.80/4.80	4.80/4.80	2.40/4.80	1.20/2.40	1.20/1.20	1.20/4.80	1.20/1.20	0.60/0.60	4.80/>19.20
Sw	0.80/0.80	1.60/1.60	1.60/6.40	0.20/3.20	0.80/1.60	0.80/0.80	1.60/6.40	0.80/6.40	0.80/1.60	>25.60/>25.60
Herbicides (mg mL^−1^)										
S met	15.00/60.00	30.00/30.00	30.00/60.00	1.87/30.00	7.50/30.00	15.00/60.00	7.50/30.00	15.00/60.00	30.00/60.00	>60.00/>60.00
Fl	>20.00/>20.00	20.00/20.00	>20.00/>20.00	20.00/20.00	20.00/20.00	20.00/>20.00	10.00/>20.00	20.00/>20.00	>20.00/>20.00	>20.00/>20.00
Insecticides (mg mL^−1^)										
Del	>0.40/>0.40	>0.40/>0.40	>0.40/>0.40	0.40/>0.40	>0.40/>0.40	>0.40/>0.40	>0.40/>0.40	0.40/>0.40	>0.40/>0.40	>0.40/>0.40
Cyp	>3.20/>3.20	>3.20/>3.20	>3.20/>3.20	3.20/>3.20	>3.20/>3.20	>3.20/>3.20	>3.20/>3.20	>3.20/>3.20	>3.20/>3.20	>3.20/>3.20

MIC/MBC—minimum inhibitory concentration/minimum bactericidal concentration; Chl—chloramphenicol (0.05–100.00 µg mL^−1^); Pen—penicillin (0.05–100.00 µg mL^−1^); Cip—ciprofloxacin (0.05–100.00 µg mL^−1^); Van—vancomycin (0.05–100.00 µg mL^−1^); Mn—MnCl_2_ × 4H_2_O (0.20–500.00 mM); Pb—PbCl_2_ (0.20–500.00 mM); Cd—CdCl_2_ (0.02–50.00 mM); Bl—Blauvit^®^ (0.06–128.00 mg mL^−1^); Eq—Equation Pro^®^ (0.01–19.20 mg mL^−1^); Sw—Swich^®^ (0.01–25.60 mg mL^−1^); S met—S metolachlor (0.03–60 mg mL^−1^); Fl—fluazifop (0.01–20.00 mg mL^−1^); Del—deltamethrin (0.0002–0.40 mg mL^−1^); Cyp—cypermethrin (0.002–3.2 mg mL^−1^). Experiment was performed in triplicate, with the same values obtained.

**Table 4 plants-14-01874-t004:** Macroelemental and microelemental contents in dried leaves of maize, tomato, cucumber, and pepper after 35 days of cultivation in greenhouse.

		Treatments										
		C	T1	T2	T3	T4	T5	T6	T7	T8	T9	T10
Macroelements (mg g^−1^ DW)												
Ca	maize	6.29 ±0.01 ^fg^	5.48 ±0.05 ^b^	5.59 ±0.04 ^c^	6.24 ±0.01 ^f^	5.99 ±0.02 ^e^	6.33 ±0.02 ^g^	5.68 ±0.01 ^d^	5.47 ±0.00 ^b^	5.72 ±0.00 ^d^	5.22 ±0.00 ^a^	5.42 ±0.00 ^b^
	tomato	21.87 ±0.11 ^d^	21.17 ±0.27 ^c^	21.12 ±0.07 ^c^	23.14 ±0.01 ^f^	22.85 ±0.00 ^ef^	22.95 ±0.03 ^ef^	22.62 ±0.21 ^e^	20.42 ±0.06 ^b^	20.37 ±0.12 ^b^	22.09 ±0.06 ^d^	20.01 ±0.03 ^a^
	cucumber	17.96 ±0.15 ^e^	16.00 ±0.05 ^b^	15.85 ±0.01 ^b^	16.63 ±0.03 ^c^	15.38 ±0.02 ^a^	19.79 ±0.18 ^g^	17.66 ±0.03 ^de^	20.45 ±0.30 ^h^	15.40 ±0.02 ^a^	19.23 ±0.07 ^f^	17.43 ±0.10 ^d^
	pepper	10.90 ±0.034 ^b^	11.57 ±0.11 ^d^	10.95 ±0.13 ^bc^	12.52 ±0.12 ^g^	11.70 ±0.06 ^d^	12.28 ±0.03 ^f^	12.57 ±0.03 ^g^	12.02 ±0.02 ^e^	11.89 ±0.01 ^e^	10.42 ±0.03 ^a^	11.10 ±0.05 ^c^
K	maize	49.53 ±0.00 ^h^	51.56 ±0.24 ^i^	47.81 ±0.09 ^g^	46.75 ±0.04 ^f^	48.04 ±0.12 ^g^	46.51 ±0.05 ^f^	43.61 ±0.04 ^d^	45.94 ±0.18 ^e^	41.82 ±0.02 ^b^	39.82 ±0.01 ^a^	42.84 ±0.11 ^c^
	tomato	34.64 ±0.28 ^b^	31.59 ±0.10 ^a^	38.22 ±0.22 ^f^	35.59 ±0.04 ^c^	36.91 ±0.25 ^de^	35.59 ±0.03 ^c^	34.78 ±0.10 ^b^	36.52 ±0.00 ^d^	37.24 ±0.18 ^e^	39.03 ±0.08 ^g^	38.39 ±0.02 ^f^
	cucumber	33.70 ±0.23 ^a^	38.43 ±0.06 ^e^	38.72 ±0.04 ^e^	37.91 ±0.00 ^d^	41.68 ±0.05 ^g^	36.78 ±0.09 ^c^	38.61 ±0.01 ^e^	39.61 ±0.29 ^f^	36.24 ±0.03 ^b^	36.95 ±0.02 ^c^	38.41 ±0.13 ^e^
	pepper	55.30 ±0.23 ^g^	54.30 ±0.32 ^f^	48.46 ±0.14 ^bc^	48.86 ±0.23 ^c^	50.10 ±0.15 ^d^	50.67 ±0.13 ^d^	52.10 ±0.26 ^e^	50.59 ±0.07 ^d^	50.31 ±0.12 ^d^	44.15 ±0.11 ^a^	48.18 ±0.29 ^b^
Mg	maize	1.93 ±0.01 ^h^	1.77 ±0.02 ^bc^	1.82 ±0.00 ^efg^	1.94 ±0.00 ^h^	1.79 ±0.00 ^cde^	1.78 ±0.02 ^cd^	1.83 ±0.00 ^fg^	1.75 ±0.00 ^b^	1.83 ±0.00 ^g^	1.71 ±0.01 ^a^	1.80 ±0.00 ^def^
	tomato	3.39 ±0.01 ^bc^	3.35 ±0.01 ^ab^	3.33 ±0.05 ^a^	3.47 ±0.00 ^d^	3.47 ±0.01 ^de^	3.57 ±0.01 ^f^	3.53 ±0.01 ^ef^	3.39 ±0.01 ^bc^	3.33 ±0.00 ^a^	3.41 ±0.01 ^c^	3.39 ±0.02 ^bc^
	cucumber	3.02 ±0.01 ^d^	2.78 ±0.00 ^a^	2.96 ±0.01 ^c^	2.93 ±0.02 ^c^	2.78 ±0.03 ^a^	3.17 ±0.01 ^f^	2.97 ±0.01 ^cd^	3.25 ±0.03 ^g^	2.78 ±0.01 ^a^	3.07 ±0.02 ^e^	2.88 ±0.01 ^b^
	pepper	3.20 ±0.01 ^i^	2.63 ±0.00 ^ef^	2.44 ±0.00 ^a^	2.50 ±0.01 ^c^	2.58 ±0.00 ^d^	2.64 ±0.00 ^fg^	2.62 ±0.00 ^ef^	2.80 ±0.00 ^h^	2.61 ±0.00 ^e^	2.48 ±0.00 ^b^	2.65 ±0.00 ^g^
P	maize	9.32 ±0.00 ^abc^	9.64 ±0.00 ^abc^	8.76 ±0.00 ^a^	10.18 ±1.27 ^c^	9.90 ±0.01 ^bc^	9.21 ±0.00 ^abc^	8.72 ±0.02 ^a^	9.19 ±0.00 ^abc^	9.43 ±0.02 ^abc^	9.04 ±0.00 ^ab^	8.92 ±0.01 ^ab^
	tomato	5.88 ±0.01 ^c^	5.53 ±0.01 ^a^	6.06 ±0.01 ^d^	5.80 ±0.00 ^b^	6.22 ±0.01 ^e^	6.32 ±0.00 ^g^	6.26 ±0.00 ^f^	6.54 ±0.00 ^i^	6.49 ±0.01 ^h^	6.71 ±0.00 ^j^	6.48 ±0.01 ^h^
	cucumber	5.56 ±0.01 ^a^	6.36 ±0.01 ^g^	6.09 ±0.01 ^c^	6.23 ±00 ^e^	6.28 ±0.00 ^f^	6.14 ±0.01 ^d^	6.14 ±0.00 ^d^	6.02 ±0.01 ^b^	6.50 ±0.02 ^i^	6.45 ±0.00 ^h^	6.59 ±0.01 ^j^
	pepper	3.65 ±0.00 ^a^	3.89 ±0.00 ^c^	4.12 ±0.00 ^e^	4.17 ±0.00 ^f^	4.34 ±0.00 ^h^	4.39 ±0.00 ^i^	4.32 ±0.00 ^g^	4.48 ±0.00 ^k^	4.45 ±0.00 ^j^	3.80 ±0.00 ^b^	3.94 ±0.00 ^d^
S	maize	3.43 ±0.00 ^e^	3.28 ±0.00 ^b^	3.27 ±0.00 ^b^	3.48 ±0.00 ^g^	3.44 ±0.00 ^e^	3.41 ±0.00 ^d^	3.34 ±0.01 ^c^	3.12 ±0.00 ^a^	3.49 ±0.01 ^h^	3.45 ±0.00 ^f^	3.56 ±0.00 ^i^
	tomato	4.64 ±0.02 ^c^	4.97 ±0.02 ^e^	4.48 ±0.00 ^b^	5.66 ±0.00 ^i^	5.43 ±0.01 ^h^	5.86 ±0.00 ^k^	5.69 ±0.01 ^j^	4.70 ±0.00 ^d^	5.03 ±0.01 ^f^	5.40 ±0.00 ^g^	4.43 ±0.01 ^a^
	cucumber	8.40 ±0.01 ^a^	8.46 ±0.00 ^b^	8.95 ±0.01 ^g^	8.97 ±0.00 ^g^	8.73 ±0.01 ^d^	8.89 ±0.01 ^e^	8.56 ±0.00 ^c^	8.49 ±0.01 ^b^	8.55 ±0.02 ^c^	8.90 ±0.00 ^f^	8.88 ±0.00 ^e^
	pepper	4.75 ±0.00 ^h^	4.68 ±0.00 ^g^	4.65 ±0.00 ^f^	5.02 ±0.00 ^k^	4.97 ±0.00 ^j^	4.85 ±0.00 ^i^	4.58 ±0.00 ^e^	4.53 ±0.00 ^d^	4.51 ±0.00 ^c^	3.99 ±0.00 ^a^	4.32 ±0.00 ^b^
Microelements (µg g^−1^ DW)												
B	maize	14.93 ±0.01 ^d^	10.34 ±0.01 ^a^	11.27 ±0.00 ^b^	13.85 ±0.65 ^c^	15.97 ±0.03 ^e^	18.51 ±0.03 ^f^	13.73 ±0.07 ^c^	13.83 ±0.03 ^c^	15.93 ±0.02 ^e^	15.47 ±0.02 ^de^	13.31 ±0.02 ^c^
	tomato	34.00 ±0.03 ^h^	30.34 ±0.08 ^c^	28.78 ±0.02 ^b^	30.30 ±0.01 ^c^	31.02 ±0.05 ^d^	33.31 ±0.01 ^g^	31.31 ±0.02 ^e^	28.70 ±0.05 ^b^	28.71 ±0.04 ^b^	32.23 ±0.01 ^f^	28.43 ±0.11 ^a^
	cucumber	34.88 ±0.03 ^g^	33.11 ±0.03 ^d^	33.99 ±003 ^e^	33.89 ±0.03 ^e^	32.00 ±0.10 ^b^	33.93 ±0.02 ^e^	32.86 ±0.04 ^c^	36.26 ±0.02 ^h^	31.53 ±0.11 ^a^	34.38 ±0.01 ^f^	31.66 ±0.01 ^a^
	pepper	36.67 ±0.01 ^e^	37.90 ±0.09 ^f^	38.10 ±0.05 ^g^	39.49 ±0.00 ^h^	41.56 ±0.01 ^i^	38.13 ±0.02 ^g^	34.55 ±0.02 ^d^	37.92 ±0.04 ^f^	34.00 ±0.01 ^c^	30.52 ±0.01 ^a^	31.30 ±0.03 ^b^
Cu	maize	9.56 ±0.01 ^a^	9.93 ±0.00 ^c^	9.69 ±0.00 ^b^	11.73 ±0.01 ^g^	10.30 ±0.03 ^e^	10.19 ±0.02 ^d^	10.35 ±0.01 ^e^	9.72 ±0.00 ^b^	11.84 ±0.03 ^h^	11.11 ±0.02 ^f^	12.52 ±0.03 ^i^
	tomato	12.41 ±0.09 ^d^	16.00 ±0.06 ^g^	16.04 ±0.08 ^g^	8.15 ±0.00 ^a^	12.59 ±0.12 ^d^	9.79 ±0.07 ^b^	10.53 ±0.09 ^c^	14.23 ±0.02 ^f^	13.45 ±0.10 ^e^	12.62 ±0.23 ^d^	12.55 ±0.12 ^d^
	cucumber	16.34 ±0.04 ^d^	15.72 ±0.06 ^b^	16.19 ±0.00 ^c^	18.25 ±0.03 ^i^	18.41 ±0.04 ^j^	16.35 ±0.01 ^d^	15.10 ±0.03 ^a^	17.00 ±0.02 ^e^	17.34 ±0.03 ^f^	17.90 ±0.07 ^h^	17.57 ±0.01 ^g^
	pepper	4. 70 ±0.06 ^e^	4.40 ±0.02 ^c^	4.35 ±0.02 ^c^	4.72 ±0.02 ^e^	4.65 ±0.01 ^de^	4.41 ±0.01 ^c^	3.74 ±0.01 ^a^	4.59 ±0.00 ^d^	3.79 ±0.01 ^a^	3.93 ±0.01 ^b^	3.87 ±0.02 ^b^
Fe	maize	77.57 ±0.05 ^b^	83.62 ±0.10 ^g^	78.48 ±0.07 ^c^	94.92 ±0.07 ^k^	83.06 ±0.19 ^f^	81.60 ±0.12 ^e^	84.48 ±0.00 ^h^	73.82 ±0.25 ^a^	90.46 ±0.05 ^i^	80.61 ±0.12 ^d^	92.15 ±0.03 ^j^
	tomato	178.82 ±0.51 ^i^	155.23 ±0.30 ^e^	139.75 ±0.42 ^a^	151.93 ±0.14 ^c^	161.79 ±0.03 ^g^	149.02 ±0.14 ^b^	172.51 ±0.47 ^h^	155.34 ±0.26 ^e^	159.49 ±0.09 ^f^	153.55 ±0.23 ^d^	155.30 ±0.43 ^e^
	cucumber	128.86 ±0.85 ^i^	115.71 ±0.07 ^f^	126.94 ±0.10 ^h^	125.02 ±0.14 ^g^	107.55 ±0.29 ^b^	133.05 ±0.52 ^j^	110.98 ±0.00 ^d^	99.18 ±0.33 ^a^	110.25 ±0.21 ^cd^	112.35 ±0.14 ^e^	109.77 ±0.38 ^c^
	pepper	74.63 ±0.34 ^a^	80.45 ±0.60 ^b^	89.18 ±0.05 ^g^	94.91 ±0.24 ^h^	87.87 ±0.17 ^f^	86.21 ±0.09 ^e^	82.20 ±0.00 ^c^	100.44 ±0.17 ^i^	85.37 ±0.14 ^d^	80.42 ±0.16 ^b^	82.07 ±0.34 ^b^
Mn	maize	128.42 ±0.14 ^c^	126.50 ±0.10 ^b^	125.77 ±0.09 ^b^	139.30 ±0.93 ^f^	143.85 ±0.26 ^g^	145.77 ±0.33 ^h^	135.37 ±0.17 ^d^	120.95 ±0.09 ^a^	137.40 ±1.33 ^e^	122.08 ±0.05 ^a^	129.16 ±0.10 ^c^
	tomato	55.90 ±1.46 ^b^	85.55 ±1.13 ^g^	71.95 ±1.14 ^f^	55.79 ±0.43 ^b^	62.27 ±1.45 ^cd^	65.71 ±0.57 ^e^	61.76 ±0.74 ^cd^	62.11 ±0.38 ^cd^	64.31 ±0.30 ^de^	59.76 ±0.42 ^c^	41.95 ±0.33 ^a^
	cucumber	111.41 ±0.66 ^i^	81.40 ±0.03 ^a^	86.78 ±0.03 ^d^	104.28 ±0.29 ^h^	100.00 ±0.29 ^g^	93.71 ±0.24 ^e^	81.64 ±0.00 ^a^	83.89 ±0.28 ^c^	82.88 ±0.12 ^b^	99.11 ±0.03 ^f^	93.68 ±0.24 ^e^
	pepper	60.26 ±0.23 ^a^	103.54 ±0.77 ^g^	112.36 ±0.05 ^j^	108.45 ±0.41 ^i^	103.73 ±0.12 ^g^	107.39 ±0.06 ^h^	80.78 ±0.17 ^f^	76.31 ±0.03 ^e^	74.01 ±0.07 ^d^	69.29 ±0.09 ^b^	70.47 ±0.32 ^c^
Zn	maize	43.99 ±0.01 ^a^	50.01 ±0.03 ^e^	49.87 ±0.03 ^d^	53.20 ±0.00 ^g^	55.03 ±0.03 ^i^	60.88 ±0.05 ^k^	46.94 ±0.05 ^c^	53.07 ±0.03 ^f^	57.80 ±0.07 ^j^	54.74 ±0.05 ^h^	45.40 ±0.03 ^b^
	tomato	67.01 ±0.12 ^e^	69.70 ±0.09 ^i^	69.16 ±0.00 ^h^	64.21 ±0.07 ^b^	66.21 ±0.09 ^d^	65.43 ±0.00 ^c^	67.24 ±0.05 ^f^	67.26 ±0.03 ^f^	67.53 ±0.07 ^g^	64.15 ±0.05 ^b^	63.49 ±0.00 ^a^
	cucumber	72.92 ±0.13 ^c^	78.86 ±0.00 ^g^	79.88 ±0.05 ^h^	82.49 ±0.03 ^j^	74.33 ±0.11 ^d^	76.72 ±0.00 ^f^	72.29 ±0.05 ^b^	71.16 ±0.07 ^a^	75.24 ±0.18 ^e^	81.02 ±0.05 ^i^	82.48 ±0.03 ^j^
	pepper	47.15 ±0.05 ^c^	50.59 ±0.12 ^g^	48.37 ±0.00 ^e^	60.28 ±0.09 ^i^	53.63 ±0.03 ^h^	49.66 ±0.02 ^f^	44.65 ±0.05 ^a^	47.98 ±0.02 ^d^	46.53 ±0.05 ^b^	47.13 ±0.03 ^c^	49.69 ±0.05 ^f^

Results are expressed as means ± standard deviations from three replicates. Different letters within each row indicate statistically significant differences among treatments (ANOVA, Tukey’s test, *p* < 0.05).

## Data Availability

The original data presented in the study are openly available in the NCBI database under the following GeneBank accession numbers. For the SUB15314772 project for the 16S rRNA gene [PV631135 (SCF5); PV631136 (SCF8); PV631137 (SCF10); PV631138 (SCF4); and PV631139 (SCF2)]; for the BankIt2959996 project for the *rpoB* gene [PV649830 (SCF9)]; and for the BankIt2960019 project for the *tuf* gene [PV649831 (SCF1); PV649832 (SCF3); PV649833 (SCF6), and PV649834 (SCF7)].

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
