# Peer review of "Growth-Promoting Effects of Ten Soil Bacterial Strains on Maize, Tomato, Cucumber, and Pepper Under Greenhouse Conditions"

_plants, 2025, doi:10.3390/plants14121874_

Round 1
Reviewer 1 Report
Comments and Suggestions for Authors
The research is scientific. However, some cogent points needed to be addressed.
Comments:
1) The title of the MS should be rephrase to capture the statement problem and the objectives of the study.
2) The values indicating the increase or decrease in the plant growth parameters should be indicated in the abstract.
3) It is not always advisable to use words in the title of a research as keywords, hence, rewrite the keywords.
4) The authors should be specific on the title whether the research is on soil bacteria or rhizobacteria.
5) The results in Tables 2 and 4 should be presented as mean±standard deviation.
6) The results in Figure 1 not reflected in the abstract. Also, only vertical and horizontal lines should show.
7) L438 and 449: The detailed methodology for each test should be clearly described.
8) L121: Enzyme screening, is it a biochemical characterization? Please clarify or rephrase the sentence.
9) There is a need for English language checks throughout the manuscript.
Minor comments.
L3: … plant growth-promoting screening…
L100: The information provided “sequencing several genes” were not provided in the MS. More details should be added to the methodology on how these genes were detected and add to the results section.
L111-112: Did authors confirm the pathogenicity of Enterobacter cloacae in their research for this affirmative statement?
L122: Can bacteria be the source of pectinase?
L140: indole
Comments on the Quality of English LanguageImprovement is required.
Author Response
Thank you for the valuable comments and constructive suggestions, which have helped us to improve the quality and clarity of the manuscript. Below, we provide a point-by-point response to each comment. All changes have been incorporated into the revised manuscript and marked using Track Changes.
Comment 1: The title of the MS should be rephrase to capture the statement problem and the objectives of the study.
Response: Thank you for the comment. We propose a new title for the MS that aligns with the statement of the problem and the objectives of the study.
Lines 6-9: Growth-Promoting Effects of Ten Beneficial Soil Bacterial Strains on Maize, Tomato, Cucumber, and Pepper under Greenhouse Conditions for Sustainable and Eco-Friendly Agriculture
Comment 2: The values indicating the increase or decrease in the plant growth parameters should be indicated in the abstract.
Response: Thank you for the comment. The most significant results regarding the increase or decrease of plant growth parameters have been included in the abstract. Lines 27-28
Comment 3: It is not always advisable to use words in the title of a research as keywords, hence, rewrite the keywords.
Response: Considering that we have changed the title of the paper, and based on your suggestion, the keywords have been modified as follows:
Lines 35-36: Keywords: identification, characterization, seed inoculation, plant morphological parameters, pigment content, elemental content
Comment 4: The authors should be specific on the title whether the research is on soil bacteria or rhizobacteria.
Response: Thank you for the comment. We have included the word 'soil' in the title of the MS.
Comment 5: The results in Tables 2 and 4 should be presented as mean±standard deviation.
Response: Thank you for the comment. All quantitative results in Tables 2 and 4 are presented as mean ± standard deviation, with statistical analysis performed in Table 4. Table 2 also shows the results of several qualitative tests, with the meaning of the labels used explained below the table.
Comment 6: The results in Figure 1 not reflected in the abstract. Also, only vertical and horizontal lines should show.
Response: Thank you for the comment. The most significant results regarding antifungal activity have been included in the abstract (Lines 30-32).
We were not entirely sure what specific changes the reviewer intended for Figure 1. Since modifying it could make it inconsistent with the style of the other figures, we decided to leave Figure 1 unchanged.
Comment 7: L438 and 449: The detailed methodology for each test should be clearly described.
Response: Thank you for the comment. In response to your suggestions, the methodology for determining the biochemical and plant growth-promoting (PGP) characteristics of the bacterial strains has been clarified in the revised manuscript.
Lines 525-547: 4.3.2. Biochemical Characteristics and Enzyme Production
Lines 551-578: 4.3.3. Plant Growth-Promotion Characteristics
Comment 8: L121: Enzyme screening, is it a biochemical characterization? Please clarify or rephrase the sentence.
Response: Thank you for the comment. We have incorporated the corrections in MS to improve clarity.
Line 137 2.2. Physiological, Biochemical and Enzymatic Characteristics of Bacterial Isolates -Enzymatic has been added to the title
Lines 138-139 and 145-158: This section of the MS has been revised to separate the enzymatic from the biochemical characterization of the isolates.
Comment 9: There is a need for English language checks throughout the manuscript.
Response: Thank you for the comment. We reviewed the English language throughout the MS and incorporated all corrections we considered necessary to improve clarity and readability.
Minor comments.
Comment L3: …plant growth-promoting screening…
Response: Thank you for the comment. As the manuscript title has been revised, this word is no longer included in the title.
Comment L100 : The information provided “sequencing several genes” were not provided in the MS. More details should be added to the methodology on how these genes were detected and add to the results section.
Response: Thank you for the comment. Although the information is contained in Materials and Methods, section 4.2, additional details are added as per your instructions. A brief explanation has also been added in the results section.
Lines 117-118: 1. Introduction
Lines 126-129: 2.1. Molecular Identification of Bacterial Isolates
Lines 490-493 and 500-505: 4.2. Molecular Identification of Bacterial Isolates
Comment L111-112: Did authors confirm the pathogenicity of Enterobacter cloacae in their research for this affirmative statement?
Response: Thank you for the comment. Pathogenicity does not need to be confirmed. E. cloacae, as the reviewer knows, could be an opportunistic pathogen and belongs to biosafety level 2 according to the French and German (according to TRBA 466) risk group classification [Ref.: #118312 and #2550, BacDive]. As we are the group dealing with strain authorization and registration, we know reliably that this strain cannot be registered in the future. A brief explanation has also been added in the results section (Lines 132-134).
Comment L122: Can bacteria be the source of pectinase?
Response: Thank you for the comment. Pectinase-producing bacteria are commonly found in environments rich in decaying plant material, where they contribute to the decomposition of plant tissues. We have provided a reference to support this statement:
Shrestha, S., Khatiwada, J. R., Zhang, X., Chio, C., Kognou, A. L. M., Chen, F., Han, S., Chen, X., & Qin, W. (2021). Screening and Molecular Identification of Novel Pectinolytic Bacteria from Forest Soil. Fermentation, 7(1), 40. https://doi.org/10.3390/fermentation7010040
Comment L140: indole
Response: Thank you for pointing out this mistake.

Reviewer 2 Report
Comments and Suggestions for Authors
The manuscript is devoted to the current topic of developing environmentally friendly and cost-effective biofertilizers based on PGPR. The authors have done a lot of experimental work. But I have several questions and comments that I believe will help improve the manuscript.
Section 2.1. Molecular Identification of Bacterial Isolates
This section of the manuscript requires careful editing. The authors should explain which "few genes" were sequenced and why they were chosen to identify the isolates. This is especially important for such a large genus as Bacillus. Unfortunately, the data in Table 1 do not allow us to assess this. The primers used would be more logically moved to the Materials and Methods section. Table 1 should indicate the combinations of genes based on the sequences used for identification. Why are the sequences of reference strains given in the table, and not the isolates under study? How can the level of homology between sequences be assessed if there is no information on the sequenced sequences of the isolates under study? The sequence data should be deposited in accessible databases.
Section 2.3. PGP Characteristics and Antifungal Activity of Bacterial Isolates
Table 2 How can the authors explain the difference in the detection of phosphate solubilization by the two tests used? Based on the identified bacterial activity presented in the table, how highly do the authors rate the growth-stimulating potential of the strains used?
Section 2.4. Sensitivity of Bacterial Isolates to Antibiotics, Toxic Metals, and Pesticides
Could the authors explain in what base solution the effect of antimicrobial agents on bacteria was determined? How was the viability of bacteria determined after the action of antimicrobial agents? Unfortunately, the methodology (reference 108) is not available. Is the resistance to heavy metals found in the studied strains comparable to the resistance of representatives of the same bacterial species?
Section 2.5. Effect of Inoculants on the Plant Growth
The authors have done a large amount of experimental work. But a detailed analysis of the measurements presented requires a more detailed discussion of the data obtained. For a number of experiments, there is no obvious correlation between the determination of the raw and dry weight of plants. Perhaps this can be due to a change in the water balance in the plants. In my opinion, this is very important, since the experiment used plants with very different physiological characteristics.
Section 2.6. Pigment Content in Leaves
How can the authors explain the lack of detectable amounts of chlorophyll b? In some cases, changes in the content of the main pigments of photosynthesis during bacterial inoculation are associated with the oxidative stress they induce. The authors believe that there is no similar effect on plants from inoculation in their work.
Table 4 contains a large array of data that is hardly discussed in the article.
I would like to see in the article a more detailed discussion of the authors' own results.
Author Response
Thank you for the valuable comments and constructive suggestions, which have helped us to improve the quality and clarity of the manuscript. Below, we provide a point-by-point response to each comment. All changes have been incorporated into the revised manuscript and marked using Track Changes.
The manuscript is devoted to the current topic of developing environmentally friendly and cost-effective biofertilizers based on PGPR. The authors have done a lot of experimental work. But I have several questions and comments that I believe will help improve the manuscript.
Response: Thank you for the comment. We appreciate your recognition of our efforts to carry out this study in the best possible way.
Comment 1: Section 2.1. Molecular Identification of Bacterial Isolates
This section of the manuscript requires careful editing. The authors should explain which "few genes" were sequenced and why they were chosen to identify the isolates. This is especially important for such a large genus as Bacillus. Unfortunately, the data in Table 1 do not allow us to assess this. The primers used would be more logically moved to the Materials and Methods section. Table 1 should indicate the combinations of genes based on the sequences used for identification. Why are the sequences of reference strains given in the table, and not the isolates under study? How can the level of homology between sequences be assessed if there is no information on the sequenced sequences of the isolates under study? The sequence data should be deposited in accessible databases.
Response: Thank you for the comments. The authors have already described in Section 4.2 which genes and why they are used for identification. This is mentioned again in the results section. Some of the authors have more than 15 years of experience in isolation and molecular identification of natural isolates (https://orcid.org/0000-0002-0425-5938), especially in the context of Bacillus identification. The tuf gene is known worldwide for the differentiation of Bacillus-related species. Detailed primer sequences have already been listed in the Material and Methods section, and neither the sequences of the primers nor those of the strains are listed in the table. Table 1 lists the genes that were used for final identification according to homology with the closest reference, and other strains with the highest similarity were retrieved from NCBI and used for comparative analysis. This is described in the M&M section. Therefore, the statements “Why are the sequences of reference strains given in the table, and not the isolates under study? &How can the level of homology between sequences be assessed if there is no information on the sequenced sequences of the isolates under study?” are simply not true. The sequences of our strains are deposited in the NCBI database, and a Data Availability Statement is included in the manuscript (Lines 719-723).
To clarify the work methodology and interpretation of the results, we have made corrections in: Lines 126-129: Section 2.1 Molecular Identification of Bacterial Isolates
Table 1: We changed “Homology to the reference strains” to “Homology to the closest reference and other strains”.
Lines 490-493 and 500-505: Section 4.2 Molecular Identification of Bacterial Isolates
Comment 2: Section 2.3. PGP Characteristics and Antifungal Activity of Bacterial Isolates
Table 2 How can the authors explain the difference in the detection of phosphate solubilization by the two tests used?
Response: Thank you for this interesting comment. The difference in the detection of phosphate solubilization between the two tests used in the study arises from the inherent characteristics of each method, and an explanation has been incorporated into the Discussion section, as suggested (Lines 333-339). We have provided references in MS to support this statement:
Pikovskaya, R; Pikovskaya, R.I. Mobilization of phosphorus in soil connection with the vital activity of some microbial species. Microbiology 1948, 17, 362-370.
Nautiyal, C.S. An efficient microbiological growth medium for screening phosphate solubilizing microorganisms. FEMS Microbiol. Lett. 1999, 170, 265-270.
Comment 3: Based on the identified bacterial activity presented in the table, how highly do the authors rate the growth-stimulating potential of the strains used?
Response: Thank you for the comment. Strains with multiple plant growth-promoting (PGP) traits are considered good candidates for use as bioinoculants, as confirmed for Bacillus safensis SCF6 (ammonia production, phosphate solubilization, HCN, IAA), Pseudomonas putida SCF9 (ammonia production, HCN, ACC deaminase activity, biocontrol), and Enterobacter cloacae SCF10 (ammonia production, phosphate solubilization, HCN, siderophore production, biofilm formation, IAA). Additionally, Bacillus subtilis SCF1 showed strong antifungal activity along with ammonia and siderophore production.
Comment 4: Section 2.4. Sensitivity of Bacterial Isolates to Antibiotics, Toxic Metals, and Pesticides
Could the authors explain in what base solution the effect of antimicrobial agents on bacteria was determined? How was the viability of bacteria determined after the action of antimicrobial agents? Unfortunately, the methodology (reference 108) is not available.
Response: Thank you for the comment. We have incorporated the details you requested in Section 4.3.5, 'Sensitivity of Bacterial Isolates to Antibiotics, Toxic Metals, and Pesticides' (Lines 618-624) to clarify the methodology.
Reference 108 has been replaced with the following reference:
Wiegand, I., Hilpert, K. Hancock, R. Agar and broth dilution methods to determine the minimal inhibitory concentration (MIC) of antimicrobial substances. Nat. Protoc. 2008, 3, 163–175.
Comment 5: Is the resistance to heavy metals found in the studied strains comparable to the resistance of representatives of the same bacterial species?
Response: Thank you for this valuable comment. We have added a sentence to the Discussion section (Lines 369-370) in which we compare the resistance to heavy metals observed in our results with published data for representatives of the same bacterial species. Since the E. cloacae strain exhibited the highest resistance to cadmium, the following reference was added as supporting evidence from previous studies:
Banerjee, G., Pandey, S., Ray, A.K. et al. Bioremediation of Heavy Metals by a Novel Bacterial Strain Enterobacter cloacae and Its Antioxidant Enzyme Activity, Flocculant Production, and Protein Expression in Presence of Lead, Cadmium, and Nickel. Water Air Soil Pollut. 2015, 226, 1-9. https://doi.org/10.1007/s11270-015-2359-9
Comment 6: Section 2.5. Effect of Inoculants on the Plant Growth
The authors have done a large amount of experimental work. But a detailed analysis of the measurements presented requires a more detailed discussion of the data obtained. For a number of experiments, there is no obvious correlation between the determination of the raw and dry weight of plants. Perhaps this can be due to a change in the water balance in the plants. In my opinion, this is very important, since the experiment used plants with very different physiological characteristics.
Response: Thank you for this valuable and interesting comment. We have added the following text to the Discussion section (Lines 404-414), addressing the correlation between the determination of raw and dry plant weight.
Comment 7: Section 2.6. Pigment Content in Leaves
How can the authors explain the lack of detectable amounts of chlorophyll b? In some cases, changes in the content of the main pigments of photosynthesis during bacterial inoculation are associated with the oxidative stress they induce. The authors believe that there is no similar effect on plants from inoculation in their work.
Response: Thank you for this valuable and interesting comment. Unfortunately, we did not perform biochemical tests to determine whether bacterial inoculation induced oxidative stress in plants. However, this is a valuable suggestion and presents a promising direction for future research. To help interpret the results related to pigment content, we have added the following text to the Discussion section (Lines 437-444):
The reduced content of chlorophyll b observed in certain maize and pepper treatments may be attributed to oxidative stress, which increases the production of reactive oxygen species (ROS) capable of disrupting enzymes involved in chlorophyll b biosynthesis (B.C. Tripathy, R. Oelmüller, 2012). Inoculation with PGPB generally enhances a plant's antioxidant capacity, but in some cases it may result in elevated levels of oxidative stress markers (Motamedi, M., Zahedi, M., Karimmojeni, H. et al., 2022). Additionally, the method used for pigment extraction plays a crucial role in determining the accuracy of chlorophyll measurements. Variables such as solvent type, temperature, light exposure, and sample preparation techniques can significantly impact pigment stability and yield (Pompelli, 2013).
B.C. Tripathy, R. Oelmüller Reactive oxygen species generation and signaling in plants Plant Signal. Behav., 7 (2012), pp. 1621-1633
Motamedi, M., Zahedi, M., Karimmojeni, H. et al. Effect of rhizosphere bacteria on antioxidant enzymes and some biochemical characteristics of Medicago sativa L. subjected to herbicide stress. Acta Physiol Plant 44, 84 (2022). https://doi.org/10.1007/s11738-022-03423-5
Pompelli, M.F.; França, S.C.; Tigre, R.C.; De Oliveira, M.T.; Sacilot, M.; Pereira, E.C. Spectrophotometric determinations of chloroplastidic pigments in acetone, ethanol and dimethylsulphoxide. Rev. Bras. Biol. 2013, 11, 52-58.
Comment 8: Table 4 contains a large array of data that is hardly discussed in the article.
Response: Thank you for this valuable comment. We have included the following text in the Discussion section (Lines 451-467), where we analyze and interpret the results presented in Table 4.
Comment 9: I would like to see in the article a more detailed discussion of the authors' own results.
Response: Thank you for this valuable comment. We hope that the changes we made based on your suggestions have improved the manuscript and that we have provided a more thorough discussion of our results

Round 2
Reviewer 1 Report
Comments and Suggestions for Authors
Revisions received. The manuscript can be accepted.
Author Response
Thank you for your positive comments.
Reviewer 2 Report
Comments and Suggestions for Authors
In my opinion, the authors have substantially edited the manuscript. I am satisfied with the authors' responses to my questions, comments and suggestions. However, I must note that the Data Availability Statement was missing from the original manuscript.
Author Response
Thank you for your positive comments.